# Leveraging SE(3) Equivariance for Self-Supervised Category-Level Object Pose Estimation

**Xiaolong Li**
Virginia Tech
lxiaol9@vt.edu

**Yijia Weng**
Peking University
halfsummer11@gmail.com

**Li Yi**
Tsinghua University
ericyi0124@gmail.com

**Leonidas Guibas**
Stanford University
guibas@cs.stanford.edu

**A. Lynn Abbott**
Virginia Tech
abbott@vt.edu

**Shuran Song**
Columbia University
shurans@cs.columbia.edu

**He Wang**[†]
Peking University
hewang@pku.edu.cn

## Abstract

Category-level object pose estimation aims to find 6D object poses of previously unseen object instances from known categories without access to object CAD models. To reduce the huge amount of pose annotations needed for category-level learning, we propose for the first time a self-supervised learning framework to estimate category-level 6D object pose from single 3D point clouds. During training, our method assumes no ground-truth pose annotations, no CAD models, and no multi-view supervision. The key to our method is to disentangle shape and pose through an invariant shape reconstruction module and an equivariant pose estimation module, empowered by SE(3) equivariant point cloud networks. The invariant shape reconstruction module learns to perform aligned reconstructions, yielding a category-level reference frame without using any annotations. In addition, the equivariant pose estimation module achieves category-level pose estimation accuracy that is comparable to some fully supervised methods. Extensive experiments demonstrate the effectiveness of our approach on both complete and partial depth point clouds from the ModelNet40 benchmark, and on real depth point clouds from the NOCS-REAL 275 dataset. The project page with code and visualizations can be found at: dragonlong.github.io/equi-pose.

## 1 Introduction

Object pose estimation is a crucial computer vision task that is widely employed in robotics, human-object interaction, and augmented-reality applications. Previous work can be categorized broadly into two genres: classic instance-level 6D object pose estimation (*e.g.*, PoseCNN [25]), which assumes the availability of exact CAD models for all object instances; and category-level 6D object pose estimation (*e.g.*, NOCS [23]), which defines a category-level reference frame and is able to generalize to a category of object instances. The motivation underlying the category-level approach is to develop systems that can accommodate novel object instances that were unseen during training. This approach is therefore more applicable for general robotic vision systems (*e.g.*, of a home robot) that must face complex environments, such as those encountered in everyday situations.

---

[†] Corresponding author.

35th Conference on Neural Information Processing Systems (NeurIPS 2021).

To achieve the intra-category generalization, category-level 6D object pose estimation systems often require a significantly larger amount of annotated training data than classic instance-level systems. The reason for more training examples lies in the need to cover large intra-category shape variations among different object instances, in addition to the usual requirement for diversity in object poses, occlusion patterns, and environments. However, obtaining good 6D pose annotations is expensive and time-consuming. As a result, this line of research often relies heavily on synthetic training data. In order to scale up category-level object pose estimation systems to allow for more object categories, there is a strong demand for self-supervised methods that do not require pose annotations.

Designing self-supervised methods for category-level pose estimation is very challenging. Although a few systems (*e.g.*, Self6D [22]) have recently been proposed for self-supervised instance-level 6D object pose estimation, all of those systems have access to the CAD model of each instance during training (but without the pose annotations). These previous approaches are thus inapplicable to self-supervised category-level object pose estimation, for which no CAD model is available for any object instances during either training or testing. The lack of any CAD models and annotations further presents a formidable obstacle to the designer: *the reference frame to define category-level pose is unspecified throughout the training*. To the best of our knowledge, there is no existing work on fully self-supervised category-level pose estimation, probably due to this issue.

We therefore need to find a way to allow the emergence of a category-level reference frame during training. A starting point is to consider the category-level reference frame known as Normalized Object Coordinate Space (NOCS), which is manually defined in [23]. Within the NOCS regression framework, all shapes are aligned in this space; also, dense canonical reconstruction of the visible points of the object of interest is performed. The relationship between the reference frame and canonical reconstruction inspires us to ask the following question: can a system learn to perform canonical reconstruction of object instances from the same object category, whose reconstruction space will naturally be a category-level reference frame? This canonical reconstruction presents two requirements: 1) all of the object reconstructions need to be aligned, and 2) the reconstruction should not change if the input object undergoes a change in pose. The second requirement suggests an SE(3)-invariant reconstruction of an observed object, which inspires us to consider point cloud processing networks that are SE(3)-invariant/equivariant.

Formally, given a set of point cloud transformations $T_A : \mathbb{R}^{N \times 3} \to \mathbb{R}^{N \times 3}$ for $A \in$ SE(3), a neural network $\phi : \mathbb{R}^{N \times 3} \to \mathcal{F}$ (where $\mathcal{F}$ is an arbitrary feature domain) is called equivariant if for each $A$ there exists an equivariant transformation $S_A : \mathcal{F} \to \mathcal{F}$ so that:

$$S_A[\phi(X)] = \phi(T_A[X]), \; \forall A \in \text{SE(3)}, \; X \in \mathbb{R}^{N \times 3}$$

Note that invariance is a special case of equivariance; the network is called invariant under the transformations when $S_A$ is an identity mapping.

Therefore, we propose to use an SE(3)-invariant network for shape reconstruction. To find a self-supervision for this reconstruction task, we propose to jointly estimate the object pose $P$ and use the predicted transformation in $P$ to transform our reconstructions into the camera space so that a consistency loss can be formed between the proposed reconstruction and our input observation. In contrast to the pose invariance pursued by our shape reconstruction module, here the estimated 6D pose should naturally be equivariant with any SE(3) transformation applied to the input, which means that the estimated pose should undergo the same 6D transformation when the input object changes its 6D pose. We thus use an SE(3)-equivariant network for this pose estimation. Leveraging SE(3) invariance and equivariance, our proposed system is designed to disentangle shape and pose in a self-supervised manner. Note that we do not explicitly enforce shape alignment in reconstruction. In other words, our network has the freedom to choose canonical poses of each instance for reconstruction during training. Even so, we find that our network automatically chooses to align all the instances for reconstruction (except for certain symmetry ambiguities), because the network is relatively "lazy" and such reconstruction is the easiest thing to do. Within this aligned reconstruction space, a category-level reference frame emerges without any supervision, which further defines our category-level pose that is to be estimated by our equivariant pose estimation module.

Extensive experiments show that this self-supervised disentanglement is successful for both complete and partial object observations. Our proposed method achieves very accurate 6D pose estimation on synthetic point clouds generated from ModelNet [24], and it works well on the real-world NOCS-REAL275 dataset [23]. An ablation study further shows the key role of equivariance and invariance in this self-supervised framework.

## 2 Related Work

**Supervised Category-Level Object Pose Estimation.**  The pioneering work of Wang et al. [23] proposed Normalized Object Coordinate Space (NOCS) as a category-specific canonical reference frame, so that the category-level pose of a previously unseen object can be defined as the transformation from its NOCS. Several follow-up efforts have improved NOCS by considering articulated objects [11], by incorporating object pose tracking [21], by leveraging analysis-by-synthesis and shape generative models [4, 6], or by exploiting learnable deformation [19]. However, all of these approaches adopt fully-supervised training paradigms and assume that object poses are known at training time. As discussed previously, object pose annotations are very difficult to obtain for category-level learning at a large scale.

**Self-supervised Pose Estimation.**  To avoid the need for pose annotations in fully-supervised learning paradigms, several works have looked into self-supervision for instance-level object pose estimation. Self6D [22] and DTPE [15] obtain object poses by minimizing the difference between rendered depth (or RGBD) images and real observations in a self-supervised manner. AAE [16] trains a pose-augmented autoencoder on synthetic images, which can then be applied to pose estimation for real images. However, these methods still assume that a corresponding CAD model is available for the target object for which pose needs to be estimated. For category-level pose estimation without any CAD models, these approaches cannot be applied. An alternative approach known as *shape co-alignment* [1, 3, 12] allows emergence of consistent category-level reference frames. These methods usually assume clean meshes as inputs, or they leverage strong priors like symmetry, making them unsuitable for more challenging shapes or even partial observations exhibiting high geometrical or topological variations. Moreover, these methods do not make the connection between shape co-alignment and category-level 6D pose estimation. QEC [27] has a problem scope closest to ours. However, instead of aiming for self-supervised category-level 6D pose estimation for both complete and partial observations, this system deals only with 3D pose (3D rotation) of complete shapes. Also, QEC does not evaluate the predicted category-level 3D pose, and there is no guarantee regarding the pose quality. Other methods like [10, 20] require multi-view RGB images as input, and the primary focus is still on shape reconstruction. In summary, our work is the first to target self-supervised category-level object 6D pose estimation. This paper demonstrates significant potential of our method using only a depth point cloud as the input.

**Equivariant Point Cloud Neural Networks.**  Recently, achieving SE(3) invariance and equivariance in point cloud processing has attracted a lot of attention. Early work [13, 26] focused more on achieving invariance, while most recent work has explored equivariance. Recent work has introduced Tensor Field Networks [18] and SE3-transformers [9], which leverage spherical harmonics; quaternion equivariant capsule networks [27]; and, most recently, Vector Neurons [7]. We note that EPN [5] is the only network for which equivariance does not entail a cost of additional network capacity.

## 3 Method

In section 3.1, we consider the Equivariant Point Network [5] with emphasis on its SE(3)-equivariance properties. Section 3.2 describes how we disentangle shape and pose, jointly estimate them, and form a self-supervised learning pipeline. Section 3.3 describes how category-level pose is inferred and evaluated at test time.

### 3.1 Revisiting Equivariant Point Networks

To achieve shape and pose disentanglement, our framework leverages a state-of-the-art SE(3) equivariant network known as the Equivariant Point Network (EPN) [5]. Taking a point cloud $X$ as input, EPN achieves equivariance regarding both 3D translation $\mathbf{t} \in \mathbb{R}^3$ and 3D rotation $g \in$ SO(3), and it can generate per-point localized equivariant features $\mathbf{f}(\mathbf{x})$ for points $\mathbf{x} \in X$. We briefly review its translation and rotation equivariance properties.

**Translation Equivariance.**  Considering 3D translation $T_{\mathbf{t}}[X] = X + \mathbf{t}$ that moves $X$ to $X'$, a translation equivariant transformation can be defined as $S_{\mathbf{t}} = I$ (identity mapping). This definition of translation equivariance is widely used in other SE(3)-equivariant networks, *e.g.*, Tensor Field

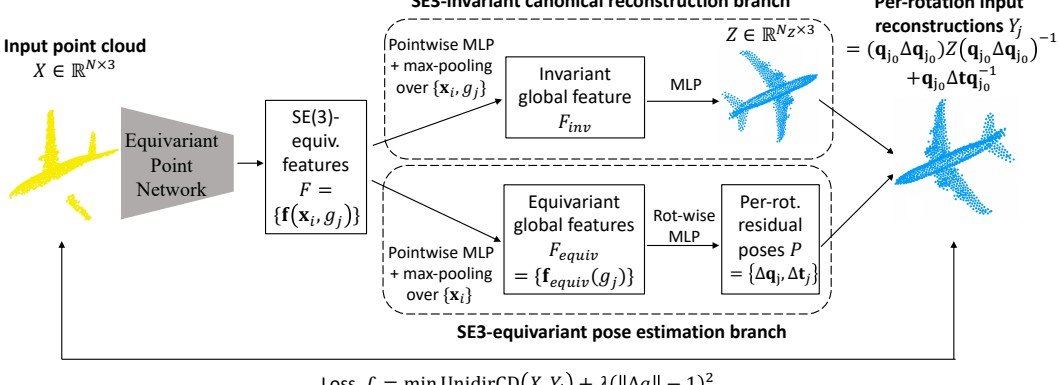

**Figure 1: Self-supervised category-level pose estimation pipeline.** Taking an input point cloud $X$ (partial or complete), our network can construct invariant canonical shape $Z$ and estimate an equivariant pose $P$, which together can form a self-supervised consistency loss to the input observation.

Network [18] and SE(3)-transformer [9]. Under this equivariance, $\mathbf{f}'(\mathbf{x}') = \mathbf{f}'(\mathbf{x}+\mathbf{t}) = S_\mathbf{t}[\mathbf{f}(\mathbf{x})] = \mathbf{f}(\mathbf{x})$, which means the feature $\mathbf{f}$ of one point $\mathbf{x}$ does not change under translation of the whole point cloud. The simplest way to produce this translation equivariance is to avoid directly processing the absolute point coordinates but instead use relative point offsets. EPN sets up local frames around each input point $\mathbf{x}$ and uses only the relative point offsets $\tilde{\mathbf{x}}_i = \mathbf{x}_i - \mathbf{x}$ during feature extraction. These point offsets can still depict the geometry but do not change under translation of the input.

Although any point network that uses only relative point offsets will be translation equivariant under this definition, EPN specifically chooses to use KPConv [17] as its convolution operator and performs convolution at each point $\mathbf{x}$ within its ball neighborhood $\mathcal{N}_\mathbf{x}$. More specifically, $(\mathbf{f} * h)(\mathbf{x}) = \sum_{\tilde{\mathbf{x}}_i + \mathbf{x} \in \mathcal{N}_\mathbf{x}} \mathbf{f}(\tilde{\mathbf{x}}_i) h(-\tilde{\mathbf{x}}_i)$, where $h(\tilde{\mathbf{x}})$ is a continuously parameterized function in $\mathcal{N}_\mathbf{x}$.

**Rotation Equivariance.** When an input point cloud is transformed by an SO(3) rotation matrix, equivariance is a desired property, especially for the sake of rotation estimation. A non-trival (non-invariant) definition of rotation equivariance thus becomes very important. The key idea behind the rotation equivariance of EPN is to introduce a feature vector field $F$ on an SO(3) manifold after discretizing the continuous SO(3) rotation group. EPN first proposes a kernel-rotated version of KPConv, which rotates the convolution kernel $h$ by a rotation $g \in$ SO(3). Using the rotated kernel $h_g$ to perform convolution on the input point cloud will result in a new set of per-point features $\mathbf{f}(\mathbf{x}, g)$, equivalent to rotating the input $X$ by $g^{-1}$. In other words, its rotated kernel function satisfies

$$h_g(\tilde{\mathbf{x}}_i) = h(T_{g^{-1}}[\tilde{\mathbf{x}}_i])$$

where $g^{-1}$ is the inverse rotation of $g$. Assume for any $g \in$ SO(3), EPN can compute per-point $C$-channel features $\mathbf{f}(\mathbf{x}, g) \in \mathbb{R}^C$ using the corresponding rotated KPConv kernel $h_g$, then the features for all $g$ and all $\mathbf{x}$ form a per-point vector field $F = \phi(X) = \{\mathbf{f}(\mathbf{x}, g) \in \mathbb{R}^C \mid g \in$ SO(3), $\mathbf{x} \in X\}$. We therefore can define a rotation equivariant transformation

$$S_{g_0}[\mathbf{f}(\mathbf{x}, g)] = \mathbf{f}(\mathbf{x}, (g_0)^{-1} \circ g).$$

Then we could prove

$$S_{g_0}[\phi(X)] = \{\mathbf{f}(\mathbf{x}, g_0^{-1} \circ g)\} = \{\mathbf{f}(T_{g_0}[\mathbf{x}], g)\} = \phi(T_{g_0}[X]).$$

Intuitively, $S_{g_0}$ will be equivalent to applying a circular shift to the vector field $F$ with the value of each entry unchanged. In practice, it is infeasible to compute $F$ for all rotations on SO(3), and EPN therefore leverages a discrete approximation and achieves equivariance for a finite subgroup of SO(3). Concretely, EPN discretizes the SO(3) rotation group into the icosahedron rotation group $G_g$, which contains 60 different rotations originating from the 60 rotational symmetries of a regular icosahedron and computes a rotated KPConv for each $g \in G_g$. This KPConv-based multi-rotation convolution is named PointConv, effectively generating a per-point 2D feature $\mathbf{f} \in \mathbb{R}^{C \times |G|}$ for all $\mathbf{x} \in X$. When

the input point cloud $X$ is rotated by some $g \in G_g$, the rotation equivariance guarantees that each **f** will be permuted in its $G$ dimension according to $g$. *I.e.*, its $j$-th feature $\mathbf{f}_j \in \mathbb{R}^C$ will become the $j'$-th feature of $\mathbf{f}'$, satisfying $g_{j'} = g \circ g_j$. To further increase the expressivity of the network and to allow information exchange between different coordinates of $F$, EPN further introduces a GroupConv operation, which passes messages across different $g$s of $\mathbf{f}(\mathbf{x}, g)$. This message passing leverages a rotation-invariant graph convolutional kernel and therefore does not break the equivariant structure of $F$. For more detailed information, please refer to EPN [5].

**SE(3)-equivariance.** By combining translation equivariance and rotation equivariance, we can define a rotation- and translation-equivariant transformation $S_A = S_\mathbf{t} \circ S_g$, so that EPN is strictly equivariant to all SE(3) transformations in $G_A = \{T_A \mid T_A[X] = T_\mathbf{t} \circ T_g[X], \forall \mathbf{t} \in \mathbb{R}^3, g \in G_g\}$. Note that $G_A$ is a subgroup of SE(3); we will discuss in section 3.2 how this subgroup approximation affects our task when the input undergoes rotations that are not in $G_g$.

## 3.2 Self-Supervised Joint Shape Reconstruction and 6D Pose Estimation

When we have only an input point cloud $X \in \mathbb{R}^{N \times 3}$ to provide supervision, jointly optimizing 3D shape and 6D pose is an ill-posed problem by nature. The network will need to jointly reconstruct an invariant canonical shape point cloud $Z \in \mathbb{R}^{N_Z \times 3}$ and estimate a 6D object pose $P = (\mathbf{q}, \mathbf{t})$, where $N$ and $N_Z$ are the number of points in $X$ and $Z$, respectively, and we choose to use quaternion $\mathbf{q}$ as our rotation representation. Here the invariant canonical shape reconstruction means the reconstruction should be invariant under all SE(3) transformations of the input point cloud. Furthermore, we want the reconstruction to be automatically aligned across different object instances under the same object category so that the reconstruction is under a category-level canonical reference frame. For equivariant 6D pose estimation, we want the estimated pose to change by the same SE(3) transformation $A$ that is applied to the input object point cloud. *I.e.*, $P(T_A[X]) = T_A[P(X)], \forall A \in$ SE(3) (approximately).

To achieve these two goals, we form a self-supervised loop by transforming the invariant canonical reconstruction $Z$ by the 6D transformation $T_P$ involved in estimated $P$ into a pose reconstruction $Y = T_P[Z]$ and enforcing consistency between input point cloud $X$ and this transformed reconstruction $Y$. We will show that, although this self-supervised loop can be formed without leveraging SE(3) equivariance, it is crucial to use a SE(3)-equivariant network for creating a category-level canonical reference, which enables category-level 6D pose estimation.

**SE(3)-equivariant Feature Backbone.** Given an input object point cloud $X$ (partial or complete), we use the EPN network as our SE(3)-equivariant backbone $\phi$ to extract per-point per-rotation SE(3)-equivariant features $F \in \mathbb{R}^{n \times C \times |G_g|} = \phi(X) = \{\mathbf{f}(\mathbf{x}_i, g_j)\}$, where $|\{\mathbf{x}_i\}| = n$ and $g_j \in G_g$. EPN comes with a stacked multi-scale encoder architecture, and will alternately perform PointConv and GroupConv in each convolution block while downsampling the points between two convolution blocks. We then feed the equivariant features over downsampled points to the following canonical shape reconstruction and 6D pose estimation branches, as shown in Figure 1.

**SE(3)-invariant Canonical Shape Reconstruction Branch.** The task of this branch is to learn an SE(3)-invariant canonical shape reconstruction function $\rho : F \in \mathbb{R}^{n \times C \times |G_g|} \to Z \in \mathbb{R}^{N_Z \times 3}$. Ideally, we want $Z$ to be invariant under arbitrary SE(3) transformations, *i.e.,* $Z(\rho(\phi(X))) = Z(\rho(\phi(T_A[X]))), \forall A \in$ SE(3). In practice, we relax this constraint and instead require $Z$ to be invariant under arbitrary transformations $A \in G_A$. To do so, we only need this learned function $\rho$ to be invariant under all transformations $S_A$ on $F$, *i.e.,* $Z = \rho(F) = \rho(S_A[F]), \forall A \in$ SE(3), thanks to the equivariance of $F$.

To implement this invariant function $\rho$, we apply an elementwise $\text{MLP}_\rho : \mathbf{f} \in \mathbb{R}^C \to \mathbf{f}_\rho \in \mathbb{R}^{C_\rho}$ to $F = \{\mathbf{f}(\mathbf{x}_i, g_j)\}$ that only operates on each $\mathbf{f}(\mathbf{x}_i, g_j)$ individually, yielding a new set of features $F_\rho = \{\mathbf{f}_\rho(\mathbf{x}_i, g_j) \mid \mathbf{f}_\rho(\mathbf{x}_i, g_j) = \text{MLP}_\rho(\mathbf{f}(\mathbf{x}_i, g_j)), \forall \mathbf{x}_i \in X', g_j \in G\}$. To obtain a global SE(3)-invariant feature $F_{inv}$, we propose to perform max-pooling over $\{\mathbf{x}_i\}$, as done in PointNet [14]. Note that for all $A \in G_A$, $S_A$ is equivalent to a permutation in the $g_j$ dimension of $F(\mathbf{x}_i, g_j)$ as well as that of $F_\rho(\mathbf{x}_i, g_j)$. Therefore we only need $F_{inv}$ to be permutation-invariant to $g_j$. This suggests another max-pooling over $\{g_j\}$. Formally, $F_{inv} = \text{max-pooling}_{i,j}(\{F_\rho(\mathbf{x}_i, g_j) \in \mathbb{R}^{C_\rho}\})$. Finally, we use a MLP-based point generator $\text{MLP}_Z$ to transform $F_{inv}$ into $\mathbb{R}^{3N_Z}$ and reshape into a point cloud $Z \in \mathbb{R}^{N_Z \times 3}$. To facilitate a canonical reconstruction, we choose to use $u() = \text{Sigmoid}() - 0.5$ as the final activation so that the reconstruction can be as zero-centered as possible.

Although $Z$ is only strictly invariant when $g \in G_g$, a rotation $g_x$ that is not in $G_g$ can be decomposed into a main component $g_{x_0} = \min_{g_j \in G_g} \text{dist}(g_j, g_x)$ and a small residual rotation $\Delta g_x$, so that $g_x = g_{x_0} \cdot \Delta g_x$. We can then see that $Z' = \rho(\phi(T_{g_x}[X])) = \rho(\phi(T_{g_{x_0}}[T_{\Delta g_x}[X]])) = \rho(\phi(T_{\Delta g_x}[X]))$. In this way, to learn an invariant shape reconstruction, our network now only needs to be invariant to rotations in the quotient group $\text{SO}(3)/G_g$, which renders a much easier learning problem. This SE(3)-invariance brings important consequences to the reconstructed point cloud $Z$: 1) if $X$ represents a complete object point cloud, then the reconstructed $Z$ will remain the same for this object under all 6D object poses; 2) for object instances with similar shapes from the same category, the network will lean towards emerging category-level aligned reconstructions, which may significantly simplify the task of the point cloud generation and resemble what happens in Quotient Network [12].

Note that when the input point cloud $X$ only depicts a partial object, as in a depth image, a pose change in the object is not equivalent to a SE(3) transformation to the point cloud due to the change in point visibility. We find in our experiments that the emerging category-level alignment still holds, but there is no guarantee that the reconstructed point clouds will be complete.

**SE(3)-equivariant 6D Pose Estimation Branch.** The task of this branch is to learn a pose estimation function $\pi$ from $F$ to an equivariant 6D object pose $P$. By saying equivariance, we mean that the estimated pose of $X' = T_A[X]$ will undergo the same 6D transformation $T_A$ that applies to the pose of the input point cloud $X$. *I.e.,* $\pi(\phi(X')) = \pi(\phi(T_A[X])) = T_A[\pi(\phi(X))] = T_A[P]$.

Inspired by the rotation estimation network introduced in EPN, we propose to predict a pose hypothesis $|G_g|$ containing one 6D pose $P_j$ for each $g_j \in G_g$. We therefore propose to learn a function $\pi : F \in \mathbb{R}^{n \times C \times |G_g|} \to \{P_j\} \in \mathbb{R}^{(3+4) \times |G_g|}$, where 3 is for translation, and 4 for a quaterion as our rotation representation. Here we parameterize each 6D pose $P_j = (\mathbf{q}_j, \mathbf{t}_j)$ using four components $(\mathbf{q}_{j_0}, \Delta \mathbf{q}_j, \mathbf{t}_0, \Delta \mathbf{t})$, explained as follows: $\mathbf{q}_{j_0}$ represents a major discrete quaternion corresponding to $g_j \in G_g$ and satisfies $g_j = \min_{q \in G_g} \text{dist}(g, g_j)$; $\Delta \mathbf{q}_j$ represents the residual quaternion, which can be composed with $\mathbf{q}_j$ to form the full quaternion $\mathbf{q}_j = \mathbf{q}_{j_0} \Delta \mathbf{q}_j$; $\mathbf{t}_0 = \bar{X}$ is the weight center of the input point cloud $X$ and is equivariant to the translation of $X$ and remains invariant for different $g$s; and $\Delta \mathbf{t}_j$ represents the residual translation under rotation $g_j$. This pose parameterization will result in $\mathbf{q}_j = \mathbf{q}_{j_0} \Delta \mathbf{q}_j$ and $\mathbf{t}_j = \mathbf{q}_{j_0} \Delta \mathbf{t} \mathbf{q}_{j_0}^{-1} + \mathbf{t}_0$, which will pose a canonical object point cloud $Z$ into $Z'_j = \mathbf{q}_j Z \mathbf{q}_j^{-1} + \mathbf{t}_j$.

To achieve pose equivariance, we adopt a similar strategy to our invariant reconstruction branch but without max-pooling over $g_j$. We learn an elementwise $\text{MLP}_\pi : \mathbb{R}^C \to \mathbb{R}^{C_\pi}$ to process each $\mathbf{f}(\mathbf{x}_i, g_j) \in F$ individually, which transforms $F$ into $F_\pi$. We then perform max-pooling on $F_\pi$ over $\{\mathbf{x}_i\}$, yielding an equivariant global feature $F_{\text{equiv}} \in \mathbb{R}^{C_\pi \times |G_g|} = \{\mathbf{f}_{\text{equiv}}(g_j) \in \mathbb{R}^{C_\pi} \mid g_j \in G_g\}$. We further learn a per-rotation $\text{MLP}_P : \mathbb{R}^{C_\pi} \to \mathbb{R}^7$ to transform each $\mathbf{f}_{\text{equiv}}(g_j)$ to a residual quaternion $\Delta \mathbf{q}_j$ and a residual translation $\Delta \mathbf{t}_j$. This pose estimation branch is equivariant with any SE(3) transformation $A \in G_A$. Similar to our argument for the invariant reconstruction, this subgroup equivariance will significantly ease the learning of pose estimation equivariance with arbitrary SE(3) transformations.

This 6D pose estimation branch has several important differences to the rotation estimation branch used in EPN. First, due to the self-supervised nature of this work, we do not know which rotation $g_j$ is the closest one to the ground truth object rotation and we therefore choose not to predict a probability distribution over $g_j$. Second, to enable 6D pose estimation, we additionally estimate a residual translation $\Delta \mathbf{t}$. Finally, EPN does not constrain the range of $\Delta \mathbf{q}_j$ because it knows which $g_j$ is the best main rotation component and can supervise the residual accordingly. Here, to avoid an overparameterization of rotation, we constrain the residual quaternion $\Delta \mathbf{q}_j$ to be a small quaternion that can only compensate for the rotation discretization introduced by $G_g$ but should not go beyond the zone of each $\mathbf{q}_{j_0}$. Note that the relationship between a quaternion $\mathbf{q} = (q_w, q_x, q_y, q_z)$ and an axis-angle $(\omega, \theta)$ allows us to constrain the rotation angle of one quaternion easily: $q_w = cos(\theta/2)$, where $\theta$ is the rotation angle. Given that the rotation angle between any two nearest-neighboring elements from icosahedron rotational group $G$ is $\pi/5$, we propose to set $\theta(\Delta \mathbf{q}) < \pi/5$ to cover the whole $SO(3)$ or $S^3$ in the quaternion space while avoding being too overparameterized. This sets a constraint to $q_w$ of $\Delta \mathbf{q}$ to be $q_w \geq \cos(\pi/10)$. Also, for a unit quaternion, $q_w \leq 1$. We can implement these two constraints via a customized activation function: $\cos(\pi/10) + (1 - \cos(\pi/10)) \cdot \text{Sigmoid}()$.

**Self-supervised Loss.** Given the canonical reconstructed point cloud $Z$ and a set of 6D pose hypotheses $\{P_j = (\mathbf{q}_j, \mathbf{t}_j)\}$, we can obtain pose $Z$ using each $P_j$, rendering posed reconstructions $Y_j = \mathbf{q}_j Z (\mathbf{q}_j)^{-1} + \mathbf{t}_j$. We then can enforce a minimum-of-$N$ loss between the input observation $X$ and the posed point clouds:

$$\mathcal{L}_{rec} = \min_{j \in \{1, \ldots, |G_g|\}} d(X, Y_j),$$

where $d : \mathbb{R}^{N_X \times 3} \times \mathbb{R}^{N_Y \times 3} \to \mathbb{R}$ represents a point cloud distance function. For complete object observation, we can choose $d$ to be bidirectional chamfer distance [8] $d_{CD}(X, Y) = \frac{1}{N_X} \sum_{\mathbf{x} \in X} \min_{\mathbf{y} \in Y} ||\mathbf{x} - \mathbf{y}||^2 + \frac{1}{N_Y} \sum_{\mathbf{y} \in Y} \min_{\mathbf{x} \in X} ||\mathbf{x} - \mathbf{y}||^2$. For partial point cloud observation, due to the incompleteness of $X$, we can only enforce a unidirectional chamfer distance $d_{\text{UnidirCD}}(X, Y) = \frac{1}{N_X} \sum_{\mathbf{x} \in X} \min_{\mathbf{y} \in Y} ||\mathbf{x} - \mathbf{y}||^2$, which only enforces each point in the partial point cloud $X$ can find a closed point in $Y$. Note that the angle constraint of $\Delta q$ implemented by constraining the range of its $q_w$ only makes sense when $\Delta q$ is a unit quaternion. We therefore additionally enforce a regularization loss $L_{reg} = (||\Delta q|| - 1)^2$. The total loss is $\mathcal{L} = \mathcal{L}_{rec} + \lambda \mathcal{L}_{reg}$, where $\lambda$ is a regularization weight.

### 3.3 Category-Level 6D Object Pose Inference and Evaluation

**Category-Level 6D Object Pose Inference.** Given an object point cloud $X$ (partial or complete), we feed it to our network and generate a reconstruction $Z$, a set of pose hypotheses $\{P_j\}$, and the corresponding posed reconstruction $\{Y_j\}$. We can then obtain its category-level 6D object pose estimation $\hat{P} = (\mathbf{q}, \mathbf{t})$ via selecting the best pose hypothesis $\hat{P} = (\mathbf{q}_{j^*}, \mathbf{t}_{j^*})$ that satisfies $j^* = \underset{j \in \{1, \ldots, |G_g|\}}{\arg\min} d(X, Y_j)$. This best pose hypothesis is the 6D object pose of the observed object instance in the reference frame of reconstruction $Z$.

**Evaluating Category-Level 6D Object Pose.** To quantitatively evaluate the performance of our self-supervised category-level 6D object pose estimation algorithm on test point cloud data $\{X_k\}$ from a given category, we need to have the ground truth category-level pose annotations $P_k^* = (\mathbf{q}_k^*, \mathbf{t}_k^*)$ for all test data $X_k$, which uniquely defines a category-level pose reference frame $R^*$.

Because our estimated poses $\hat{P}_k$ are in the reference frames $R_k$ of reconstruction $Z_k$, there can be center shift and orientation differences between $R_k$ and $R$ or even among different $R_k$s. We therefore need to find a consistent reference frame $\hat{R}$ and register it to $R^*$. We propose the following registration algorithms.

For each test data $X_k$, we transform it using the inverse transformation of $P_k^*$, yielding a canonicalized object point cloud $Z_k^* = (\mathbf{q}_k^*)^{-1} X_k \mathbf{q}_k^* - \mathbf{t}_k^*$. We then feed $Z_k^*$ to our network for pose estimation and obtain a pose misalignment $(\tilde{\mathbf{q}}_k, \tilde{\mathbf{t}}_k)$. We propose to use a RANSAC-based method to find a majority consensus of $\tilde{\mathbf{q}}$ and $\tilde{\mathbf{t}}$. Then, for all test data, our final pose prediction can be computed as $(\hat{\mathbf{q}}\tilde{\mathbf{q}}^{-1}, \ \hat{\mathbf{t}} - \hat{\mathbf{q}}\tilde{\mathbf{q}}^{-1}\tilde{\mathbf{t}}\hat{\mathbf{q}}\hat{\mathbf{q}}^{-1})$ using our network's raw pose prediction $(\hat{\mathbf{q}}, \hat{\mathbf{t}})$.

## 4 Experiments

### 4.1 Datasets and Evaluation

**Datasets.** We evaluate our method using the following datasets. 1) Complete shape point clouds from ModelNet40 [24]. We first test our method on ModelNet40 with five categories, including airplane, car, chair, sofa and bottle. Random rotation is applied to each input data during training, and 5 different rotations are randomly sampled for each test instance. We follow the original train/test split from the ModelNet40 dataset. 2) Depth point clouds of objects from ModelNet40. We evaluate our method on partial depth observations by rendering the depth images of the complete object instances described above. For depth rendering, we first put the camera at a fixed distance facing the object, and the object is allowed to have small offsets and random rotations. For each category, we generate 30K training images with 6K testing images. The dataset is made public in our project page. 3) Real-world depth point clouds from the NOCS dataset [23]. We further evaluate our algorithm on the partial object depth point clouds from the NOCS-REAL275 dataset . This dataset contains a synthetic training set with 275K discrete frames generated with 1085 object models in the classes

Table 1: **Rotation estimation on complete point cloud.** We report mean and median rotation error, and accuracy for 3D pose estimation on rotated ModelNet40 Dataset. The best performance is in **bold** and the second best is underscored.

| Mean(°)↓ / Med.(°)↓ / 5° mAP ↑ | Airplane | Car | Chair | Sofa | Bottle |
|---|---|---|---|---|---|
| EPN(supervised) | **3.35** / **1.12** / **0.96** | **9.48** / **1.85** / **0.95** | 8.56 / **3.87** / **0.68** | **4.76** / **1.56** / **0.97** | 49.72 / 43.19 / 0.03 |
| KPConv(supervised) | 14.85 / 10.78 / 0.12 | 38.16 / 19.26 / 0.06 | 20.39 / 12.01 / 0.06 | 128.62 / 134.40 / 0.00 | **1.33** / 1.36 / **1.00** |
| ICP + template shape | 8.11 / 1.22 / 0.90 | 22.76 / 2.94 / 0.69 | 88.92 / 96.28 /0.10 | 39.00 / 9.69 / 0.28 | 76.11 / 54.42 0.04 |
| Ours | 23.09 / 1.66 / 0.87 | 17.24/ 2.13 / 0.89 | **7.05** / 4.55 / 0.56 | 8.87 / 3.22 / 0.68 | 4.42 / **0.83** / 0.98 |

Table 2: **6D pose evaluation on ModelNet40 partial depth point clouds.** The evaluation metrics include mean and median rotational error, mean and median translational error, and 5° accuracy together with 5°0.05 accuracy. Oracle means the supervised. The best performance is in **bold** and the second best is underscored.

| | Mean(°)↓ / Med.(°)↓ / 5°↑ | Airplane | Car | Chair | Sofa | Bottle |
|---|---|---|---|---|---|---|
| Rotation | EPN(oracle) | 19.8 / 4.4 / 0.61 | 92.3 / 93.2 / 0.13 | **8.6** / **3.2** / **0.75** | **20.8** / **3.01** / **0.79** | 22.8 / 15.4 / 0.17 |
| | KPConv(oracle) | 11.1 / 7.4 / 0.24 | 107.8 / 113.4/ 0.00 | 26.5 / 14.6 / 0.07 | 35.8 / 16.9 / 0.04 | **6.2** / 2.0 / **0.89** |
| | ICP + template shape | 14.7 / 1.54 / 0.83 | **87.4** / **15.0** / **0.36** | 68.19 / 13.47 / 0.29 | 115.0 / 165.4 / 0.11 | 26.11 / **1.50** / 0.81 |
| | Ours | **3.31** / **1.46** / **0.95** | 124.9 / 177.3 / 0.12 | 10.4 / 3.84 / 0.62 | 72.2 / 7.9 / 0.40 | 46.8 / 2.3 / 0.71 |
| | Mean ↓ / Med. ↓ / 5° 0.05 ↑ | | | | | |
| Translation | EPN(oracle) | 0.10 / 0.09 / 0.10 | 0.27 / 0.27 / 0.00 | 0.10 / 0.09 / 0.13 | 0.09 / 0.08 / **0.15** | 0.14 / 0.11 / 0.03 |
| | KPConv(oracle) | 0.09 / 0.08 / 0.02 | 0.18 / 0.12 / 0.00 | 0.09 / 0.07 / 0.02 | **0.07** / **0.06** / 0.02 | 0.10 / 0.09 / 0.23 |
| | ICP + template shape | 0.09 / 0.08 / 0.05 | **0.04** / **0.04** / **0.34** | 0.08 / 0.07 / 0.19 | **0.07** / 0.07 / 0.04 | **0.05** / **0.04** / **0.60** |
| | Ours | **0.04** / **0.02** / **0.76** | 0.07 / 0.06 / 0.06 | **0.05** / **0.04** / **0.44** | **0.07** / 0.07 / 0.10 | 0.13 / 0.10 / 0.19 |

from ShapeNet [2] along with 7 real training videos capturing 3 different instances of objects for each category. Its testing set, NOCS-REAL275, has 3,200 frames collected from 6 real videos containing novel instances. We assume that the ground truth object mask is available in our experiments and use them to crop the objects from the depth.

**Metrics.** We perform all pose evaluation on held-out instances per category. For each category, we report rotational error $R_{err}$, and translational error $T_{err}$ in the form of mean, and median values, together with 5° and 5cm accuracy. For symmetric objects such as bottles, we evaluate the angular error of the up-axis direction.

**Baselines.** We compare against two supervised methods: EPN and KPConv, which serve as performance oracles. We utilize a tuned implementation of the ICP algorithm as a traditional unsupervised method, where we use the same set of 60 rotations from icosahedron group for rotation initializations. Because we assume that we do not have complete ground-truth shapes for testing, for ICP we randomly select an example shape, and iteratively register the observation to the example shape. Note there are no existing learning-based unsupervised methods that fit our setting, especially on partial shapes with both 3D rotation and 3D translation.

## 4.2 Experiments on Synthetic Object Point Clouds

**3D pose estimation from complete object point clouds.** We first consider rotation of complete shapes. As shown in Table 1, although our method is fully self-supervised, it achieves comparable results to supervised method using state-of-the-art point cloud learning models such as EPN and KPConv. For certain categories, our method even surpassed the supervised oracles such as the mean rotational error on chair, and median rotational error on bottle. EPN gives the best supervised performance on all the categories except for bottle, mainly due to the difficulty of rotational mode classification under rotational symmetry ambiguity. Without the equivariance property on full SO(3) space, KPConv struggles to give an accurate rotation estimation, and our self-supervised pipeline out-performs it on all 4 asymmetric categories, while performing well on symmetric bottle. For simple shapes like airplane and car, ICP algorithm could achieve a reasonable estimation after iterative alignment, but fails at chair and sofa category. It also does not work well for the bottle case as the given example shape does not match well with all different bottle instances.

**6D pose estimation from partial object depth point clouds.** When the input observation is partial, there will be more ambiguities in determining object canonical pose. Despite this challenge, we observe that our method is still able to infer well-aligned canonical shapes across different instances, as shown in Table 2 and Figure 2. For some object categories, our method is able to infer a complete

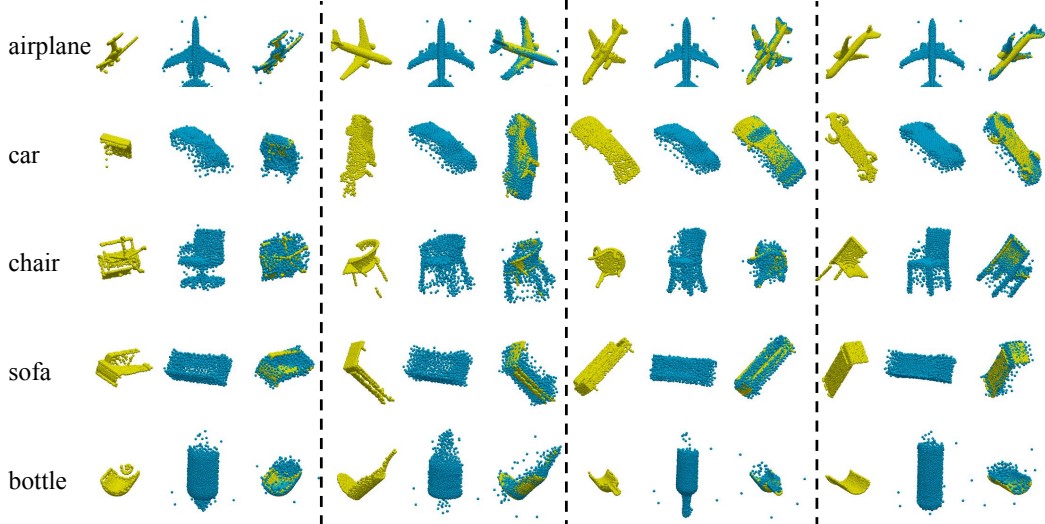

Figure 2: Visualization on the ModelNet partial shapes. Here we show four test instances for each categories: left: input $X$, middle: canonical reconstruction $Z$, and right: posed reconstruction $Y$. Note that $Z$s have been calibrated.

Table 3: **Pose estimation with equivariant and non-equivariant bakcbones.**

| Dataset | Shape | Pose | Mean $R_{err}(°)$ | Median $R_{err}(°)$ | Mean $T_{err}$ | Median $T_{err}$ | 5° | 5° 0.05 |
|---|---|---|---|---|---|---|---|---|
| Complete airplane | EPN | EPN | **14.24** | **1.53** | - | - | **0.91** | - |
| | EPN | KPConv | 21.24 | 1.70 | - | - | 0.88 | - |
| | KPConv | EPN | 133.89 | 169.96 | - | - | 0.00 | - |
| | KPConv | KPConv | 134.27 | 172.21 | - | - | 0.02 | - |
| Partial airplane | EPN | EPN | **3.47** | **1.58** | **0.02** | **0.02** | **0.95** | **0.88** |
| | EPN | KPConv | 4.24 | 1.96 | 0.03 | **0.02** | 0.93 | 0.85 |
| | KPConv | EPN | 134.34 | 174.98 | 0.07 | 0.08 | 0.05 | 0.05 |
| | KPConv | KPConv | 132.88 | 172.31 | 0.077 | 0.08 | 0.04 | 0.032 |

object shape by training on different view points. However, for bottle, the algorithm only considers the object shape as a 'half' bottle due to the symmetry – partial bottle shape can already cover all possible view points. Quantitatively, results in Table 2 show that our method achieves the best accuracy on both partial airplane and chair data. It also handles the symmetric object like bottle well, while the two supervised methods suffer from performance degradation due to the increased difficulty.

**Effects of equivariance.** As shown in Table 3, we compare the method's performance when we use equivariant or non-equivariant backbones. The results demonstrate the importance of using equivariant neural networks for canonical shape reconstruction. Leveraging SE(3)-equivariance makes it possible to generate consistently aligned canonical shapes, which in turn helps get accurate pose estimation, using non-equivariant backbone for shape would fail totally. We also show that using equivariant backbone for direct pose regression would further improve the pose estimation accuracy, especially on partial shapes.

**Different equivariant backbones.** As shown in Table 4, we compare the performance of different equivariant backbones, with both SE(3)-equivariant networks like EPN and SE(3) transformer. SE(3) transformer also promises rotational and translational equivariance by leveraging properties of spherical harmonics.From our experiments, it could also get consistently aligned canonical reconstructions. However, compared to EPN, SE(3) transformer is way more computationally heavier, and SE(3) transformer struggles to generate accurate rotation estimation compared to EPN, especially on partial shapes. EPN-20 is another variant of EPN network, when the full SO(3) space is divided into 20 different sub-regions, which has smaller number of sub-spaces than our finalized EPN-60 backbone. From our experiment it doesn't get better performance than EPN-60. Based on these experimental findings, we finally choose EPN-60 as our SE(3)-equivariant backbone.

**Failure cases.** When heavy self-occlusions and shape symmetry present, the performance of our model tends to degrade. We observe both sofa and car categories have a lot of 180-degree-flip,

Table 4: **Pose estimation with different equivariant backbones.**

| Dataset | Backbone | Mean $R_{err}(°)$ | Median $R_{err}(°)$ | Mean $T_{err}$ | Median $T_{err}$ | 5° | 5° 0.05 |
|---|---|---|---|---|---|---|---|
| Complete airplane | SE(3) Transformer | **6.65** | 3.64 | - | - | 0.7 | - |
| | EPN-20 | 64.12 | 2.40 | - | - | 0.63 | - |
| | EPN-60 | 23.09 | **1.66** | - | - | **0.87** | - |
| Partial airplane | SE(3) Transformer | 19.82 | 5.29 | 0.07 | 0.04 | 0.46 | 0.13 |
| | EPN-20 | 85.53 | 5.15 | 0.03 | 0.03 | 0.50 | 0.45 |
| | EPN-60 | **3.47** | **1.58** | **0.02** | **0.02** | **0.95** | **0.88** |

Table 5: **6D pose evaluation on NOCS-REAL 275 dataset.** Full evaluation of category-level 6D pose estimation on our ModelNet40 depth dataset using different methods. The evaluation metrics include mean and median rotational error, mean and median translational error, and 5° accuracy together with 5°5cm accuracy. The best performance is in **bold** and the second best is underscored.

| | Mean(°)↓ / Med.(°)↓ / 5°↑ . | Bottle | Bowl | Camera | Can | Laptop | Mug |
|---|---|---|---|---|---|---|---|
| Rotation | 1) | 14.8/3.0/0.79 | **5.1/3.5/0.73** | 67.8/39.5/0.06 | 2.9/2.4/0.86 | **71.0/4.1/0.55** | **7.7/4.7/0.53** |
| | 2) | 9.0/**2.6/0.86** | 5.3/5.1/0.48 | **38.0/15.3/0.07** | 2.9/2.5/0.88 | 84.8/8.1/0.39 | 13.1/5.0/0.49 |
| | 3) | 10.5/4.5/0.57 | 9.0/7.5/0.22 | 152.3/167.8/0.0 | 4.6/3.9/0.68 | 88.7/13.6/0.01 | 123.3/126.6/0.00 |
| | 4) | **7.7**/3.5/0.73 | 6.3/5.9/0.38 | 87.1/79.2/0.00 | 3.2/2.7/0.87 | 86.4/5.3/0.48 | 134.8/134.3/0.00 |
| | 5) | 56.7/22.1/0.04 | 11.0/9.9/0.20 | 119.1/115.8/0.00 | **2.7/2.3/0.90** | 148.1/178.4/0.04 | 101.1/127.3/0.00 |
| | **Mean(m)↓ / Med.(m)↓ / 5°5cm↑** | | | | | | |
| Translation | 1) | **0.04/0.03/**0.79 | **0.04/0.04/0.73** | **0.05/0.05/**0.06 | **0.05/0.04**/0.86 | 0.03/0.03/**0.55** | **0.04/**0.04**/0.53** |
| | 2) | 0.07/0.07/**0.86** | 0.05/**0.04**/0.48 | 0.08/0.08/**0.07** | 0.13/0.13/0.87 | **0.02/0.02**/0.39 | **0.04/0.03/**0.49 |
| | 3) | 0.05/0.05/0.57 | **0.04/0.04**/0.22 | 0.07/**0.06**/0.00 | 0.14/0.13/0.68 | 0.06/0.06/0.01 | 0.21/0.22/0.00 |
| | 4) | 0.05/0.05/0.73 | 0.08/0.08/0.38 | 0.07/0.07/0.00 | 0.16/0.16/**0.87** | **0.02/0.02/**0.48 | 0.12/0.12/0.00 |
| | 5) | 0.13/0.12/0.04 | 0.06/0.06/0.20 | 0.11/0.11/0.00 | 0.09/0.08/**0.90** | 0.06/0.06/0.04 | 0.09/0.07/0.00 |

which means the model couldn't distinguish dominant direction like front and back. The error of rotation estimation also contributes to the translational error for car and sofa. For the bottles, we also observe multiple cases where the reconstructed shape is upside down and an considerable offset to the dominant object center which results in the poor performance for translation estimation. See our appendix for the error percentile curves.

## 4.3 Experiments on Real-World Object Point Clouds

To test our method under a more realistic setting and explore the limits of its performance, we examine our method on the challenging real world depth data from NOCS-REAL275[23] dataset, where sensor noises and heavy occlusions in the real world cluttered scene further complicates the problem, e.g. the delicate handle structure of the mug category is often invisible, making it difficult to estimate its pose and shape. In addition, different from our randomly rotated synthetic data, this real dataset contains a biased pose distribution, e.g. bottles are always standing upright and the bottom part is never observed, adding difficulty to the reconstruction task. We consider five different experiment settings: 1) perform supervised pretraining on CAMERA[23] dataset, similar to EPN; 2) self-supervisedly finetune the pretrained weight of 1) using the small train split of NOCS; 3) self-supervisedly pretrain on CAMERA; and finally 4) self-supervisedly finetune the pretrained weight of 3) on NOCS-train, together with 5) ICP algorithm. We report a full evaluation of category-level 6D pose estimation on our NOCS-REAL275 dataset using different methods in Table 5. Our fully unsupervised method beats ICP on almost all categories and performs quite well on bottle, bowl, and can, without using any labelled or real data. Self-supervised finetuning on real data further improves its performance.

## 5 Conclusion

This paper has presented a novel self-supervised network that is capable of category-level 6D object pose estimation. The system operates without CAD models or annotations, and it achieves impressive accuracy on both synthetic and real-world category-level pose estimation benchmarks. This work is the first to demonstrate the power of SE(3)-equivariant point cloud networks for 6D pose estimation, especially on partial object point clouds. As limitations, the performance of SE(3) equivariant networks may degrade when objects exhibit symmetries, or viewpoint distribution is highly biased, or occlusions are always heavy. Extending this work to a semi-supervised setting may help to relieve these limitations, which we leave for future works. This work can reduce the demand of 6D pose annotation, therefore may impose negative impacts to people who work on object pose annotations.

**Acknowledgement:** This research is supported by a Vannevar Bush faculty fellowship, NSF grant IIS-1763268, and gifts from the Adobe and Autodesk Corporations. We also appreciate resources provided by Advanced Research Computing in the Division of Information Technology at Virginia Tech.

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
