# Leveraging SE(3) Equivariance for Self-Supervised Category-Level Object Pose Estimation Supplementary Material

**Xiaolong Li**
Virginia Tech
lxiaol9@vt.edu

**Yijia Weng**
Peking University
halfsummer11@gmail.com

**Li Yi**
Tsinghua University
ericyi0124@gmail.com

**Leonidas Guibas**
Stanford University
guibas@cs.stanford.edu

**A. Lynn Abbott**
Virginia Tech
abbott@vt.edu

**Shuran Song**
Columbia University
shurans@cs.columbia.edu

**He Wang**[†]
Peking University
hewang@pku.edu.cn

## A  Implementation details

In this section, we first provide a detailed description of our network architecture. We then describe our training protocol for experiments on different datasets. Finally, we also provide additional details about the baselines we use in our main submission.

### A.1  Network architecture

We leverage EPN as our SE(3)-equivariant backbone. The core element of EPN is the SPConv block, which consists of one SE(3) point convolution layer to learn features from spatial domain $S$, and one SE(3) group convolution operator to learn features within rotational group $G$. Batch normalization layers and leaky ReLU activations are inserted in between and after. We implement a 5-layer hierarchical convolutional network, with each layer containing two SPConv blocks. The first SPConv block is strided by a factor of 2 to down-sample input points gradually. We sample 1024 points from each input data, the points will be downsampled to 64 points after the last SPConv layers with each point containing features that are equivariant to the rotation group $G$. We set the output feature channel as 128. For the SE(3)-invariant canonical shape reconstruction branch, we implement the $\text{MLP}_Z$ as a simple 2-layer MLP to map the pooled SE(3)-invariant feature into $1024 * 3$, which corresponds to 1024 points in canonical space. The 2-layer MLP is followed by Sigmoid activation function to constrain the values. For the SE(3)-equivariant 6D pose estimation branch, $\text{MLP}_\pi$ is realized by another 2-layer MLP with batch normalization and a leaky ReLU activation. $\text{MLP}_P$ is just a one-layer MLP. In practice, the residual translation prediction head is separated from the residual rotation one, where a 2-layer MLP is used instead.

### A.2  Training protocol

We implement our framework using Pytorch 1.7.1. For all the training, we use an Adam optimizer with an initial learning rate of 0.0005, and the learning rate decay is 0.9995 per step. For all experiments on complete shapes, we train for 500 epochs over the whole dataset; for experiments on ModelNet40 partial data, we train the model for around 500k steps until convergence. For all pre-training on CAMERA-synthetic data, we let the model run more than 1000k steps until

35th Conference on Neural Information Processing Systems (NeurIPS 2021).

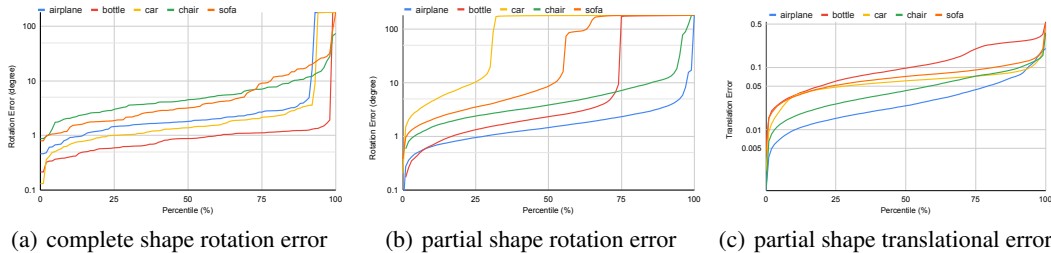

(a) complete shape rotation error  (b) partial shape rotation error  (c) partial shape translational error

Figure 1: **Error percentiles for complete and partial shapes from ModelNet40.**

convergence. We then run other 150k steps for self-supervised fine-tuning on the NOCS-REAL275 training set. We use the self-supervised reconstruction loss to train the network, the weighting factor for the additional quaternion regularization is empirically set as 0.1.

### A.3 Baselines

We train EPN and KPConv in a fully supervised manner across all our experiments for the performance comparison. Both EPN and KPConv are SOTA methods for 3D point cloud processing. More specifically, we adapt EPN-60 to have an additional 3D translation prediction head, and we adapt the classification variant of KPConv to do rotation and translation prediction at the same time. Since there are no existing methods that focus on self-supervised 6D pose estimation from depth point clouds, we mainly use ICP as an unsupervised baseline. ICP requires canonical shape for alignment optimization, here we only assume we have template shapes for each category, and we average the scores after randomly picking 3 different template shapes.

## B  Additional Results and Visualization on ModelNet40

### B.1  Error percentiles on ModelNet40

To better understand our model's performance on category-level data, we draw the error percentile curves in Figure 1. In plot (a), for all the categories, more than 75% of the data samples in test set have a rotation error smaller than $10°$. With complete shapes as input, only airplane and sofa have a small portion of cases that the rotation error is bigger than $100°$, those errors mainly corresponds to $180°$ flips. However, with partial data as input, there are more $180°$ flips cases, especially for sofa and car categories as shown in plot (b). When heavy self-occlusions and shape symmetry present, the model would usually find it difficult to get an accurate shape reconstruction. Under those cases when shape reconstruction quality is poor compared to the groundtruth, pose hypothesis with a flipped orientation estimation might get a smaller uni-direction Chamfer distance compared to the correct ones. Due to the above issue, the model will find it hard to distinguish dominant direction like front and back, or up and down for certain partial inputs. The error of rotation estimation also contributes to the translation error for car and sofa like in plot (c).

### B.2  Effects of biased viewpoints distribution

For a lot of real applications, the view points of the collected data is biased and may not cover the full SO(3) space, i.e., some regions would never be seen. This would first cause difficulty in reconstructing the amodal shape. We study the robustness of our method under biased views by limiting the sampled viewpoints from upper semi-sphere only. As shown in Table 1, the performance of our method is affected when testing on novel instances, but the performance drop is reasonable.

### B.3  Effects of dataset scale

Self-supervised learning would benefit from large-scale data, it is always an interesting question when the dataset scale is limited. Here in our case, the performance would drop consistently when

Table 1: **Effects of biased viewpoints.**

| Dataset | Viewpoints | Mean $R_{err}(°)$ | Median $R_{err}(°)$ | Mean $T_{err}$ | Median $T_{err}$ | 5° | 5° 0.05 |
|---|---|---|---|---|---|---|---|
| Partial airplane | Unbiased | **3.47** | **1.58** | **0.02** | **0.02** | **0.95** | **0.88** |
| Partial airplane | Biased | 5.63 | 2.14 | 0.03 | **0.02** | 0.91 | 0.83 |

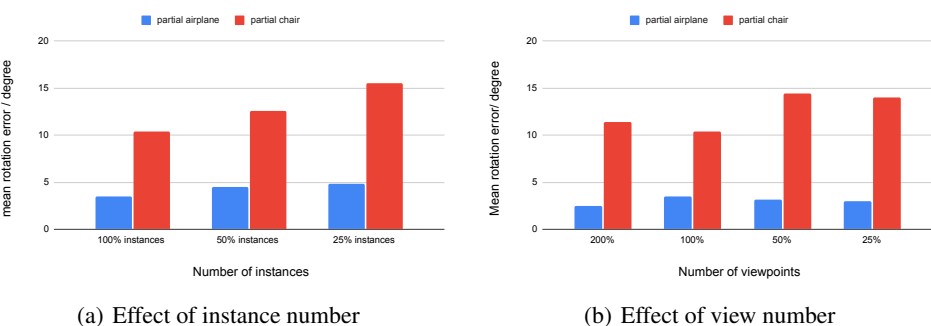

(a) Effect of instance number

(b) Effect of view number

Figure 2: **Performance under different dataset scale**

we have less number of input instances. When we reduce the input viewpoints per instance to some degree, our method's performance would also be slightly affected.

### B.4 Visualizations of Canonical Reconstruction

We first visualize our canonical shape reconstructions in Figure 3 and Figure 4. In Figure 3 we show reconstructions from rotated complete shapes, and in Figure 4 we visualize reconstructions from partial depth inputs. For experiments with partial shapes, the reconstructed shapes might not be complete as can be seen for cars and bottles. This is because the model won't see the complete object during training, and the model might perceive the shape as 'partial'. When the canonical reconstruction is not complete, the model still has the strategy to generate accurate rotation and translation estimation to match the partial input and minimize the Chamfer distance loss, as shown in Figure 7 for the bottle case.

### B.5 Self-Supervised Pose Estimation with Different Backbones

Here we shown the canonical shape reconstructions with different backbones in Figure 5.

### B.6 Visualizations of Input, Reconstruction and Pose Prediction Overlay

Here we show point cloud pairs of input, canonical reconstruction, and transformed canonical reconstruction from testing stage with ModelNet40 depth data in Figure 6 and Figure 7. Note that all the chosen instances are kept the same with Figure 4 for better reference. The canonical reconstruction task is challenging since the input is randomly rotated and translated partial shapes, and there is no supervision of groundtruth shape due to our self-supervised setting. Despite of the challenge, both the canonical reconstruction and 6D pose are well disentangled and recovered by our method.

## C   Additional Results on NOCS-REAL275 dataset

We further visualize the canonical reconstruction of method 2) in Figure 8.

## D   Self-Supervised Instance-Level Pose Estimation on YCB-Video Dataset

Our method can also be applied to self-supervised instance-level object pose estimation from depth only, in the case that the ground truth object model is given.

In this setting, we remove our reconstruction branch and replace the network-generated reconstruction $Z$ with points sampled from the object surface. We test our model on three representative objects

from the YCB-Video dataset [1]: *002_master_chef_can* with continuous symmetry, *003_cracker_box* with discrete symmetry, and *035_power_drill* without symmetry. Similar to our experiments on NOCS-REAL275 dataset, we assume ground truth object masks are provided and we only use depth point clouds as the input, which makes some textured, asymmetric objects in the original dataset textureless and symmetric in our case. Since this setting is different from most instance-level pose estimation works, we only compare to ICP-60 (described in the main paper) which runs under the same setting.

We report mean rotation error, mean translation error (using object model diagonal length as unit 1), ADD AUC [1] (for asymmetric objects), and ADD-S AUC [1] (for symmetric objects). For *002_master_chef_can*, we use the angle difference between predicted and ground truth symmetric axes for the rotation error computation. For *003_cracker_box*, we transform an object pose annotation following the object's symmetric transformations to generate all ground truth poses, and then evaluate the predicted pose error *w.r.t.* the closest ground-truth. Though trained in a fully self-supervised manner, our model produces fairly good results and outperforms ICP-60 in all cases.

Table 2: **6D pose evaluation on YCB dataset.** *ADD does not apply to *002_master_chef_can* with continuous symmetry.

| Category | Method | Mean $R_{err}(°)\downarrow$ | Mean $T_{err}(\times10^{-2})\downarrow$ | ADD AUC $\uparrow$ | ADD-S AUC $\uparrow$ |
|---|---|---|---|---|---|
| 002_master_chef_can | Ours | **1.84** | **3.20** | 23.69* | **83.47** |
| | ICP-60 | 6.10 | 4.75 | 27.58* | 75.62 |
| 003_cracker_box | Ours | **4.57** | **2.84** | **71.37** | **86.68** |
| | ICP-60 | 28.12 | 8.27 | 37.15 | 60.23 |
| 035_power_drill | Ours | **2.29** | **1.25** | **87.26** | **95.15** |
| | ICP-60 | 5.60 | 2.23 | 83.45 | 89.20 |

# E Potential Negative Social Impact

This work can reduce the demand of 6D pose annotations needed for training category-level object pose estimation and may lead to future improvements in this field. The work therefore may impose negative impacts to people who work on object pose annotations. Also, the improvements in object pose estimation may facilitate robotic research and might affect some people's job in long term.

# References

[1] Yu Xiang, Tanner Schmidt, Venkatraman Narayanan, and Dieter Fox. Posecnn: A convolutional neural network for 6d object pose estimation in cluttered scenes. *arXiv preprint arXiv:1711.00199*, 2017. 4

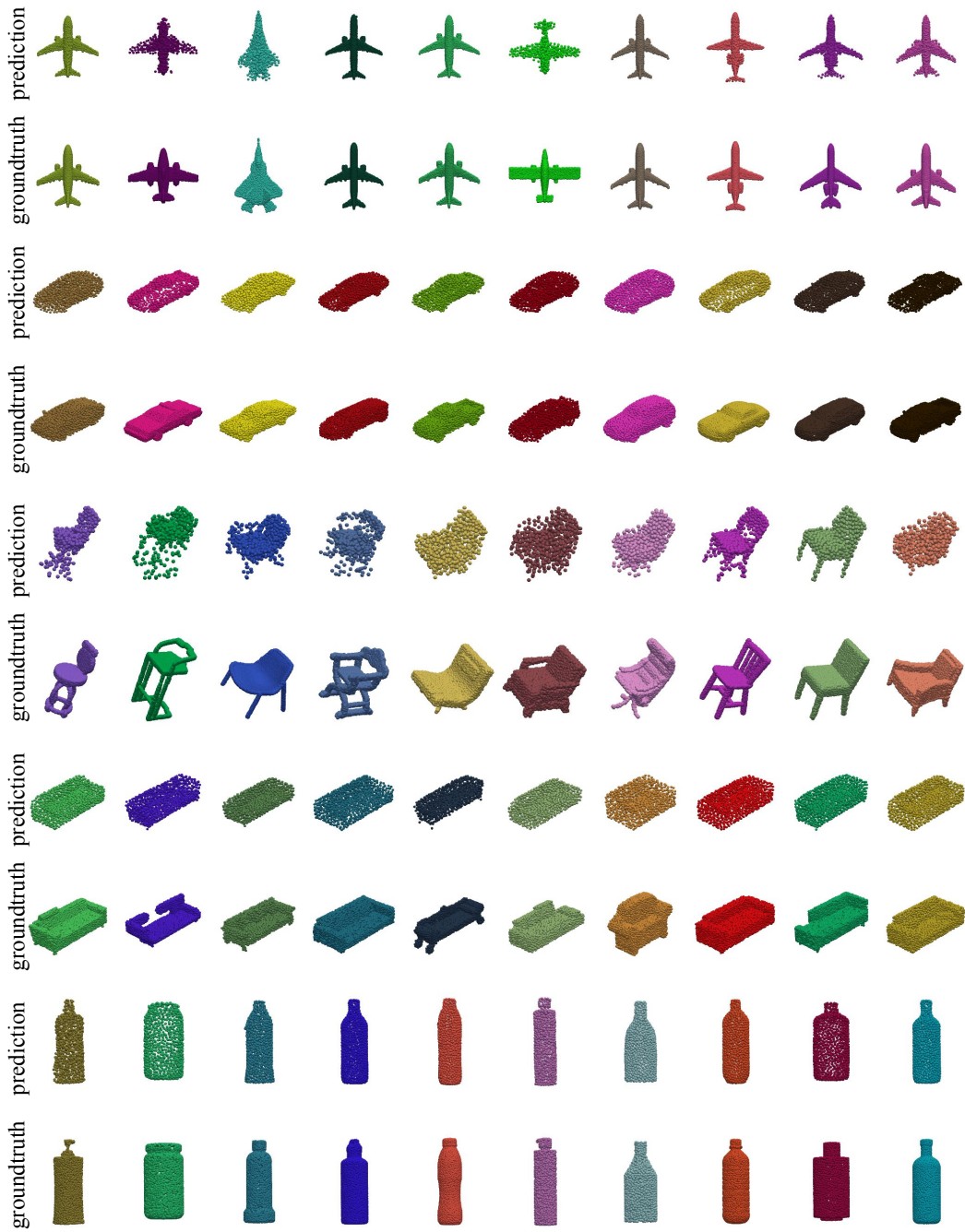

Figure 3: **Reconstruction visualization on the ModelNet complete shapes.** Here we show 10 test instances for each category, with both canonical prediction together with corresponding the groundtruth shapes. All the examples showing here are novel instances randomly chosen from test set.

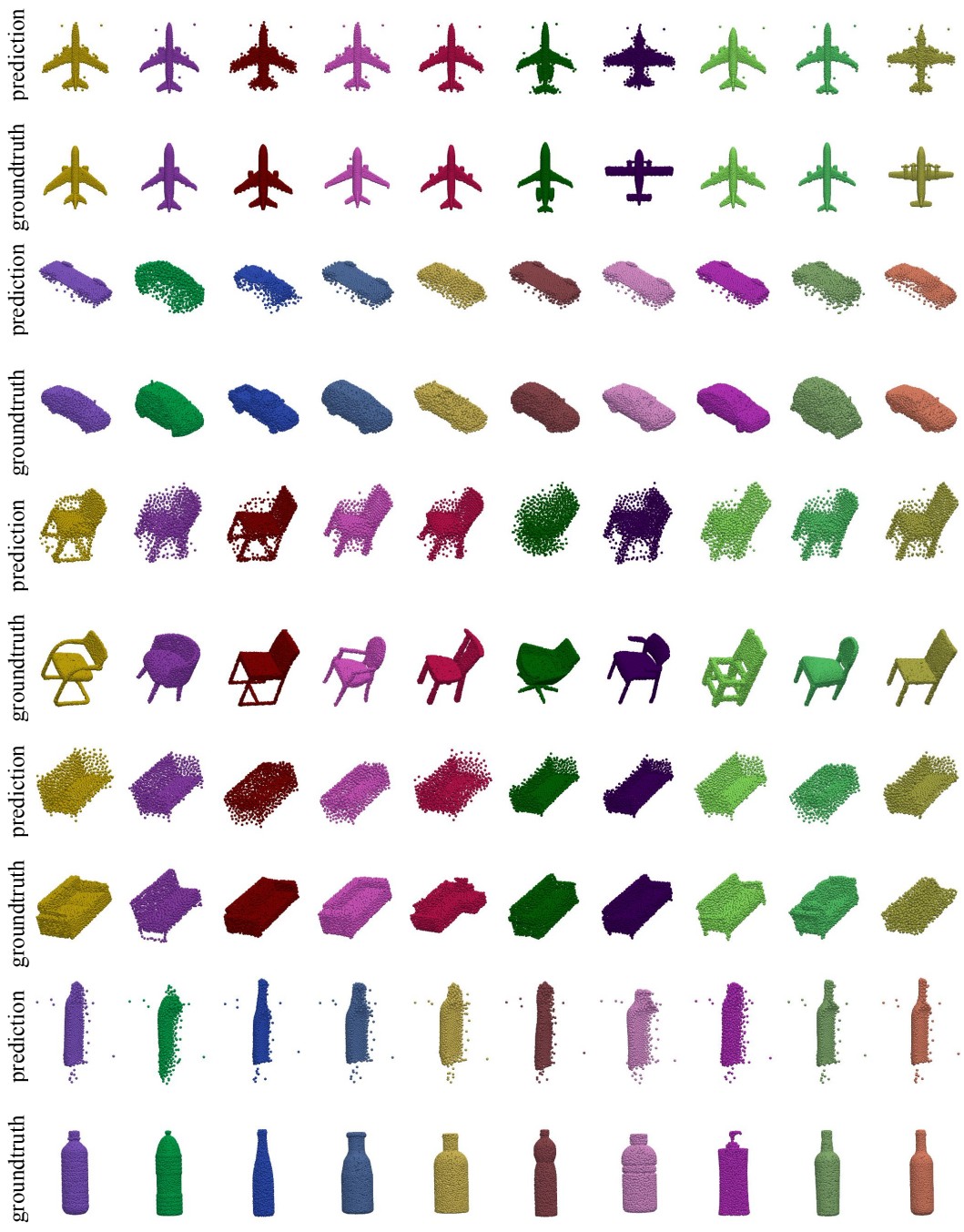

Figure 4: **Reconstruction visualization on the ModelNet partial shapes.** Here we randomly choose 10 test instances for each category and show both the calibrated canonical reconstruction and corresponding GT shapes. Please also check Figure 6 and 7 on the next pages for corresponding input and predicted pose.

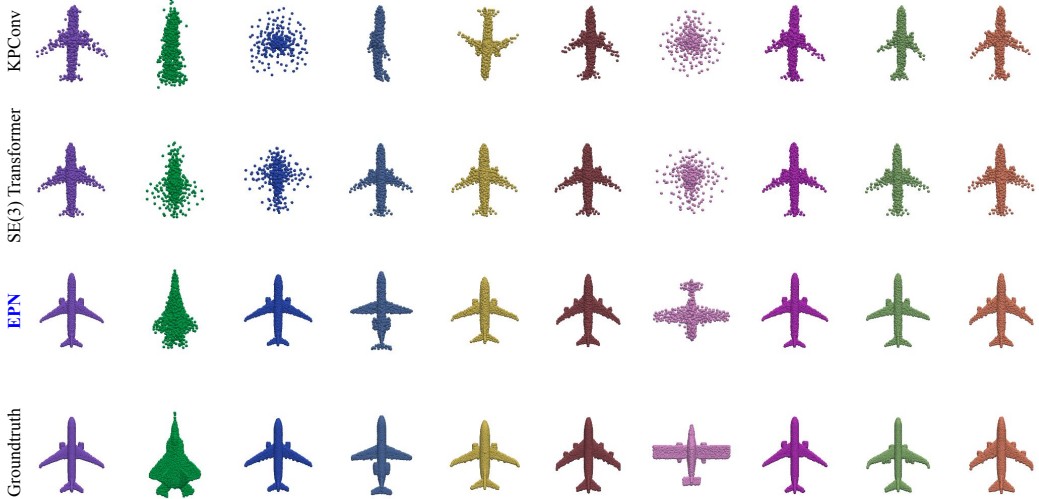

Figure 5: **Canonical reconstruction visualization with different backbones.** Here we show 10 test instances for the airplane category. All the examples showing here are randomly chosen novel instances from the test set.

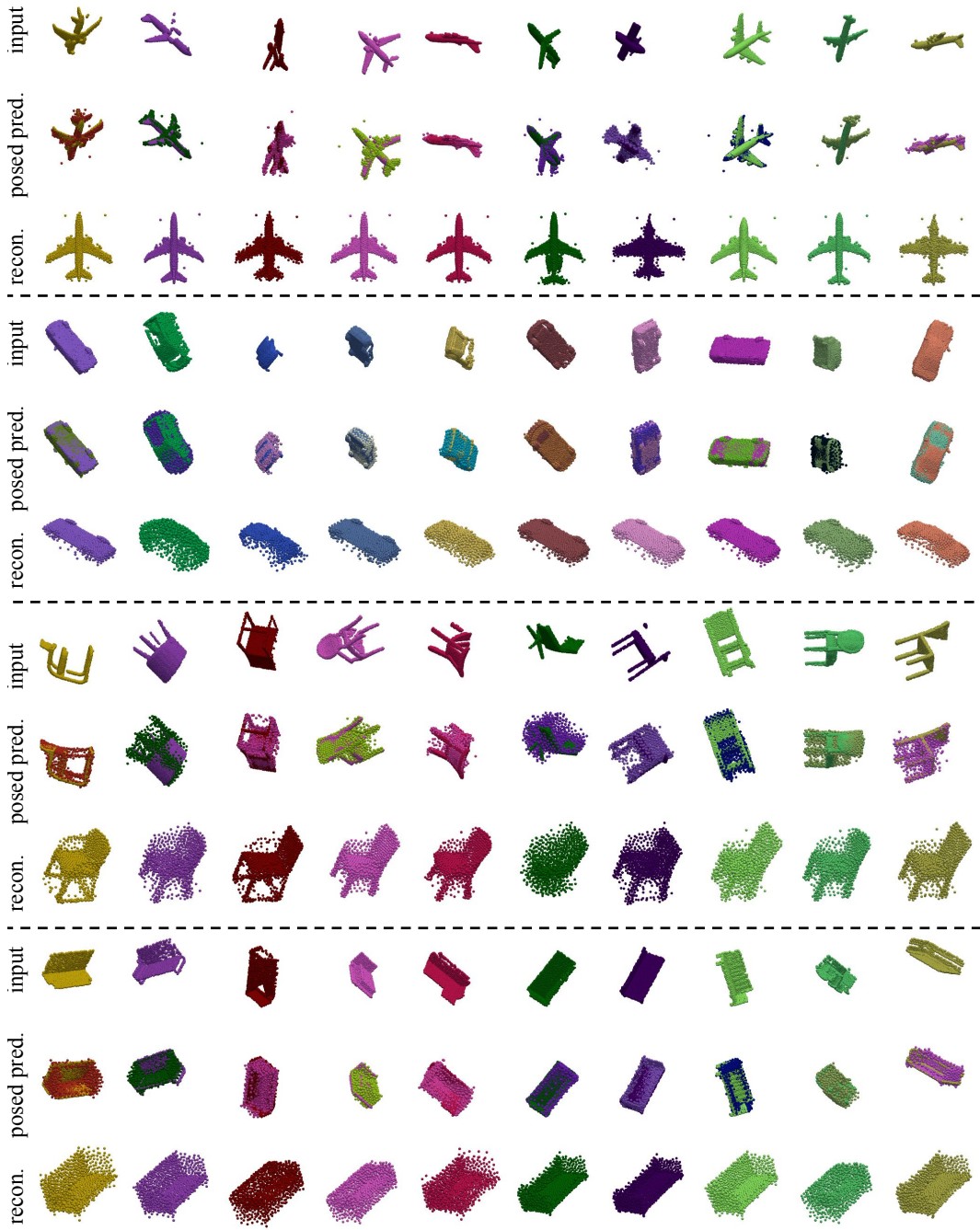

Figure 6: **6D pose estimation on the ModelNet partial shapes.** Here we show 10 test instances for each category. The predicted reconstruction is overlapped to input partial shape using the predicted pose. All the examples showing here are novel instances from the test set.

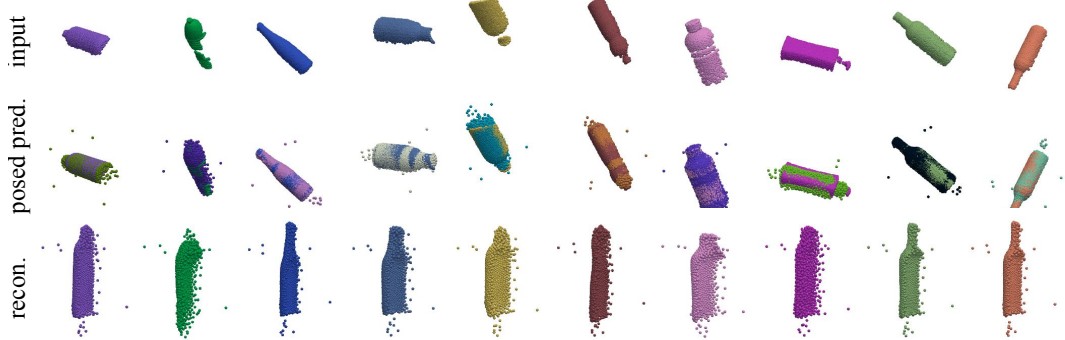

Figure 7: **6D pose estimation on the ModelNet partial bottle.** Even though it's not complete, the posed prediction still matches well with uni-directional Chamfer loss.

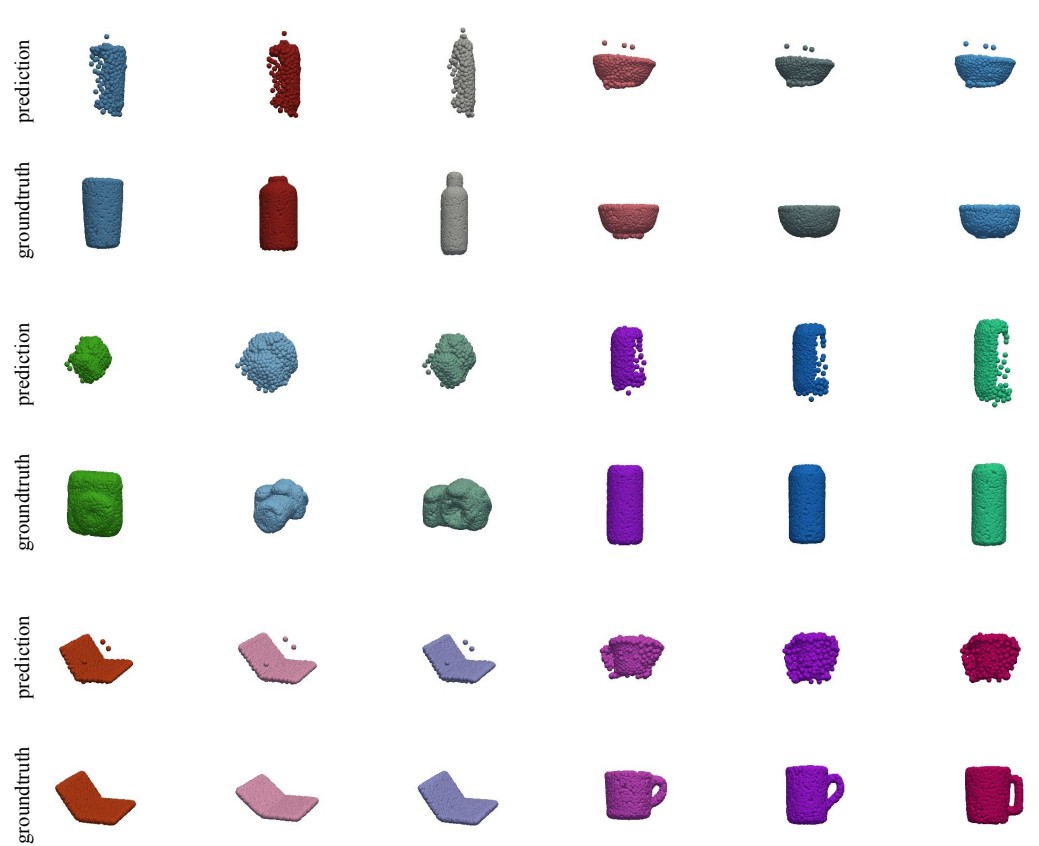

Figure 8: **Visualization of reconstruction of partial shapes from NOCS-REAL275 depth data.** Here we show randomly chosen real and novel test instances for each category, only novel partial observations from camera space are used as input.