# OpenReview forum: "Leveraging SE(3) Equivariance for Self-supervised Category-Level Object  Pose Estimation from Point Clouds"
_NeurIPS.cc/2021/Conference — NeurIPS 2021 Poster_

### Official Review · Reviewer_VRcx · 2021-07-12

**Rating:** 6
**Confidence:** 4

**Summary:**

The paper proposes a framework to learn 6D object pose estimation without GT pose annotations and CAD models (while GT object masks are available, and each object is already isolated). The key is to jointly learn canonical shapes and poses by a SE(3)-equivariance network. It shows promising performance on both synthetic and real depth data.

**Limitations And Societal Impact:**

The authors have adequately addressed the limitation.

**Main Review:**

Strongness:
- Clear writing
- Experiments on multiple datasets, including both synthetic and real ones
- Detailed analysis of failure cases and limitations

---

Weakness:
1. The authors miss some important related works. Although relying on multi-view supervision, there is a line of previous works focusing on learning-shape-from-x without GT labels, e.g. [1], [2], [3]. Especially, [3] has proposed a quite similar framework, like disentangling canonical shapes and poses, self-supervised losses, as well as selecting the best hypothesis during inference. The major differences are the type of supervision (multi-view) and the network design (PointNet + regression). It also presents many observations mentioned in this paper. For example, symmetric objects like cars are difficult for such self-supervised frameworks.
2. Although there are no existing approaches to compare directly, the authors could provide more ablation studies to verify the design choices, which is kind of missing now. Although Sec 2.4 in Supp has presented some results, can the authors analyze the impact of equivariant networks on the shape and pose separately? For example, KPConv (or EPN) for shapes and EPN (or KPConv) for poses. Compared to [3], one key improvement of this paper is to introduce SE(3)-equivariant networks to generate consistent canonical shapes. Another question is how important it is to predict multiple hypotheses for 6D poses. Can the authors study the impact of the number of hypotheses (predefined groups)? And can the authors directly regress poses rather than using any hypothesis?
3. The second car in Figure 2 looks strange. The blue point cloud (canonical shape) in the 3rd column looks different from that in the 2nd column.
4. How stable is the training process? It seems that the network can easily get stuck in local minima (imperfect shapes will lead to wrong poses, and imperfect poses can also lead to imperfect shapes. This phenomenon might be most obvious for cars). Can the authors report the result of multiple trials of experiments to show the variance of the method?

---

Typos:
- L152: there are two "j-th", and I think one should be "j'-th".
- L203-L204:  $g_x=\Delta g_{x_0} \cdot g_x$

---
References
- [1] Unsupervised Learning of Shape and Pose with Differentiable Point Clouds
- [2] Multi-view Consistency as Supervisory Signal for Learning Shape and Pose Prediction
- [3] Weakly-supervised 3d shape completion in the wild

**Time Spent Reviewing:**

3

---

> ### Author Response · Authors · 2021-08-10
> **Response to Reviewer VRcx**
>
> We thank reviewer VRcx for listing more related works and contributing critical thinkings to the experiments. We will add reference to the mentioned related works to give a better background.
>
> #### Q1: can the authors analyze the impact of equivariant networks on the shape and pose separately? More ablation studies to verify the design choices.
> To further analyze the impact of equivariant networks on the shape and pose separately, we add ablation studies on: 1. KPConv for pose + EPN for shape; 2. EPN for pose + KPConv for shape, as shown below. We found that equivariant network is the key to achieve good performance, especially on canonical shape estimation.
>
> <center>Different backbones for pose and shape separately</center>
>
> | Dataset           | Shape Backbone | Pose Backbone | Mean R_err | Median R_err | Mean T_err | Median T_err                        | 5deg. Acc. | 5deg.0.05 Acc.|
> | :---------------- | :------------- | :------------ | :--------- | :----------- | :--------- | :---------------------------------- | :------------ | :---------- |
> | Complete airplane | EPN            | EPN           | **14.24**  | **1.53**     | /          | /                                   | **0.91**          | /           |
> | Complete airplane | EPN            | KPConv        | 21.24      | 1.70         | /          | /                                   | 0.88          | /           |
> | Complete airplane | KPConv         | EPN           | 133.89     | 169.96       | /          | /                                   | 0.00          | /           |
> | Partial airplane  | EPN            | EPN           | **3.47**   | **1.58**         | **0.02**       | **0.02**                                 | **0.95**          | **0.88**        |
> | Partial airplane  | EPN            | KPConv        | 4.24       | 1.96         | 0.03       | **0.02**                                | 0.93          | 0.85        |
> | partial airplane  | KPConv         | EPN           | 134.34     | 174.98       | 0.07       | 0.08                                | 0.05          | 0.05        |
>
> #### Q2: Questions on multiple hypotheses for 6D poses. Better study on the impact of the number of hypotheses (predefined groups) or directly regress poses.
> * This ‘multiple pose hypothesis’ is determined by our EPN backbone, since each pose prediction is assigned to a subspace of the SO(3) space, only by using all of them we could cover the whole icosahedron rotation group and achieve rotational equivariance. The same MLP layer is shared to generate pose predictions per feature instead of using multiple heads.
> * In our ablation study with SE(3)-transformer backbone(which is also SE(3) equivariant), we directly regress poses instead of using multiple hypotheses, but the results are not as good as our proposed method;
> * Here we provide experiments where we set the predefined rotation group to be with 20 elements, instead of using 60. We show the results as below for both complete and partial airplane data, in which EPN60 works much better than EPN20.
>
> | Dataset           | Backbone | Mean R_err | Median R_err | Mean T_err | Median T_err                        | 5deg. Acc. | 5deg.0.05 Acc.|
> | :---------------- | :------- | :--------- | :----------- | :--------- | :---------------------------------- | :------------ | :---------- |
> | Complete airplane | EPN20    | 64.12      | 2.40         | /          | /                                   | 0.63          |/            |
> | Complete airplane | EPN60    | **14.24**  | **1.53**     | /          | /                                   | **0.91**      | /           |
> | Partial airplane  | EPN20    | 85.53      | 5.15        | 0.03        |  0.03                               |  0.50             |   0.45    |
> | Partial airplane  | EPN60    | **3.47**   | **1.58**     | **0.02**   | **0.02**                            | **0.95**          | **0.88**        |
>
> #### Q3: How stable is the training process? Result of multiple trials of experiments to show the variance of the method.
>
> Here we show multiple runs of the same experiments on car, airplane categories for complete point clouds and partial point clouds.
> * Our training process is usually stable for different categories with both complete and partial inputs, like for complete car and airplanes, also partial airplane data;
> * We do observe for trainings on car category, the performance variance is much bigger than airplane category;
> * Multiple runs on partial cars also show that our method has difficuly toward partial car data.
>
>
> | Dataset           | run_id | Mean R_err       | Median R_err   | Mean T_err | Median T_err                                                 | 5deg. Acc.  | 5deg.0.05 Acc.|
> | :---------------- | :----- | :--------------- | :------------- | :--------- | :----------------------------------------------------------- | :------------- | :---------- |
> | Complete airplane | 1/2/3  | 14.24/17.47/16.23     | 1.53/1.26/1.82      | /          | /                                                            | 0.91/0.89/0.91      | /           |
> | Complete car      | 1/2/3  | 15.46/14.16/9.95 | 2.19/1.74/1.94 | /          | /                                                            | 0.89/0.91/0.95 | /           |
> | Partial airplane  | 1/2  | 3.91/3.47     | 1.75/1.58 | 0.02/0.02  | 0.02/0.02  |  0.94/0.95  | 0.86/0.88   |
> | Partial car       | 1/2/3  | 138.19/136.51/124.86   | 176.06/176.09/177.29  | 0.11/0.05/0.07  | 0.11/0.04/0.06 | 0.18/0.17/0.12      | 0.00/0.13/0.06   |
>
> #### Q4: Strange visualization on partial car in figure 2.
> Thanks for pointing out this, we believe is due to the poor downsampling when overlaying the points for visualization. This has been fixed in our updated visualizations.

---

### Official Review · Reviewer_8FBM · 2021-07-14

**Rating:** 7
**Confidence:** 3

**Summary:**

This paper proposes a self-supervised approach to 6 DoF pose estimation. Usually such methods are either instance-level, assuming known CAD models for all object instances; or category-level, assuming a reference frame that can be generalized to all instances of a certain category. The second approach is more generalizable, however it requires a substantial amount of annotated training data, often from synthetic datasets. So far, self-supervision in this setting has been limited to instance-level, with CAD models but without pose annotations. This paper proposed category-level self-supervision and claims to be the first to do so, eliminating the need of both CAD models and large-scale annotated data.

**Ethical Concerns:**

I believe the ethical concerns of this paper is minimal and pertains to the ones inherent to self-supervised learning. For example, removing unwanted dataset biases, correct anonymization when learning from unlabeled data, ensuring the correct diversity so it includes different aspects of the data, and so forth.

**Limitations And Societal Impact:**

The authors mention certain limitations of their approach, such as degradation when objects exhibit symmetries, or viewpoint distribution is highly biased, or under heavy occlusions. There are no quantitative ablations or examples of such cases from the datasets considered in experiments, that would have been helpful to further illustrate these points. As for societal impact, the authors mention negative impact to people who work on object pose annotations.

**Main Review:**


I find this paper quite novel, and as far as the authors claim (I am not an expert in the field) this is the first method to achieve self-supervision on category-level 3D object pose estimation. The reasoning behind how the authors achieve this is also well-explained and motivated throughout the paper (although I found it difficult to read in some points, see below). This also makes the paper significant, as it opens up a lot of potential follow-up work in self-supervised learning for this another task.

Overall, the paper should be proof-read and improved in terms of grammar, presentation and clarity. Particularly the Method section, probably due to lack of space a lot is introduced in long sentences and paragraphs, it is hard to parse all the information. Having sub-section and proper equations with numbers and details would have been helpful. Same thing with experiments, there are a lot of long sentences and paragraphs with combined information that makes it hard to parse.

Other comments:

---

The contributions of this paper are three-fold:

- A general self-supervised framework for category-level 6D object pose estimation without CAD models or annotations.
- Showing that the concept of category-level reference frame can be recovered naturally during self-supervised training
- High accuracy in synthetic and real-world category-level datasets

I think contributions 1 and 2 can be combined into one, as the authors are proposing 1 by means of 2. About 3, "achieving high accuracy" is vague, it would be better to report relative improvements and other real comparisons with other methods. Is this method outperforming other self-supervised methods, or competitive with supervised methods?

---

About experiments, I would like to have seen ablations with different amounts of training data. This is the biggest benefit of using self-supervised learning, so improvement with data is an important aspect to show as a way to potentially boost results even further.

--

On table 1, why does the proposed method outperforms EPN only for chair, and why does KPConv outperforms EPN only for bottle, but by a very large margin?

--- Minor comments

line 360: Unfinished sentence.
line 117: "only one network" or "the only network"?  I didn't understand this sentence.  EPN achieves equivariance without increasing in complexity?



**Time Spent Reviewing:**

2

---

> ### Author Response · Authors · 2021-08-10
> **Response to Reviewer 8FBM**
>
> We thank the reviewer for the constructive advices, and will properly address the reviewer’s suggestions on our paper writing. We tune contribution point 3 as ‘our proposed method achieves accurate pose estimation that is comparable or surpasses existing supervised SOTA methods’.
> #### Q1: ablations with different amounts of training data.
>
> We evaluate on the same validation sets for both the airplane and the chair categories, but with different training data following the settings below:
>    - 25% or 50% data by using less instances, while keeping the same number of viewpoints;
>    - 25% or 50% data by reducing viewpoints per instance, while keeping the same number of instances;
>    - 200% data by increasing the viewpoints;
>
> We can draw several interesting conclusions from the table below.
> * **[1]** While we do expect the model performance to drop when we have smaller amount of data, the model's performance doesn't drop much when we only use 25% of the training instances for both airplane and chair categories from the ModelNet40 dataset.
> * **[2]** We also found that when only using 25% of the original number of viewpoints(15 views per instance), the model is still able to estimate the 6D poses reasonably well on the evaluation set.
> * **[3]** As expected, more viewpoints would help the model get a more accurate rotation estimation for partial inputs.
> <center>Ablation study over different amounts of training data</center>
>
> | Dataset           | Training Data   | Mean R_err | Median R_err | Mean T_err | Median T_err                        | 5deg. Acc. | 5deg.0.05 Acc. |
> | :---------------- | :-------------- | :--------- | :----------- | :--------- | :---------------------------------- | :------------ | :---------- |
> | Complete airplane | 100% instances   | **14.24**  | **1.53**     | /          | /                             | **0.91**      | /           |
> | Complete airplane | 50% instances    | 20.22      | 2.38         | /          | /                                   | 0.86          | /           |
> | Complete airplane | 25% instances    | 18.38      | 2.41         | /          | /                                   | 0.88      | /           |
> | Partial airplane  | 100% instances   | **3.47**   | 1.58         | **0.02**       | **0.02**                                | **0.95**        | **0.88**        |
> | Partial airplane  | 50% instances    | 4.53       | **1.34**         | **0.02**       | **0.02**                                | 0.94          | 0.87              |
> | Partial airplane  | 25% instances    | 4.84       | 2.02         | **0.02**       | **0.02**                                | 0.94          | 0.87              |
> | Partial airplane  | 200% viewpoints  | **2.44**    | 1.53         | **0.02**       | **0.02**                                | **0.96**        | **0.88**        |
> | Partial airplane  | 50% viewpoints   | 3.14       | **1.37**         | **0.02**       | **0.02**                                | 0.95          | **0.88**        |
> | Partial airplane  | 25% viewpoints   | 3.00       | 1.49         | **0.02**       | **0.02**                                | **0.96**        | **0.88**        |
> | Partial chair     | 100% instances   | **10.40**  | **3.84**         | **0.05**       | **0.04**                                | **0.62**        | **0.44**        |
> | Partial chair     | 50% instances    | 12.55       | 4.08        | 0.06      | 0.05                                | 0.59          | 0.39             |
> | Partial chair     | 25% instances    | 15.52       | 4.26          | 0.06      | 0.05                                | 0.57          | 0.39              |
> | Partial chair     | 200% viewpoints  | **11.41**   | **3.85**     | **0.05**       | **0.04**                                | **0.61**        | **0.41**        |
> | Partial chair     | 50% viewpoints   | 14.41       | 4.28         | 0.05       | 0.04                                | 0.57          | 0.41       |
> | Partial chair     | 25% viewpoints   | 13.99       | 4.35          | 0.06      | 0.05                                | 0.56          | 0.38              |
>
> #### Q2: why does the proposed method outperform EPN only for chair, and why does KPConv outperforms EPN only for bottles, but by a very large margin?
>
> * **[Specialness of chair category]** On table 1 for complete shapes over different categories, EPN is supposed to achieve highly accurate results due to the effective neural network design and direct pose supervision, but our results are actually comparable to EPN regarding the median error. As pointed out, our model even outforms EPN on the complete chair category regarding the mean rotation error. We conject this is relevant to the unique structure of the chair category. Exploring the category-specific synergy between the reconstruction and the pose estimation could be an interesting future work.
> * **[EPN doesn't handle well symmetric categories]** EPN is the SOTA method for 3D point clouds processing and has unique advantage on pose estimation tasks, however the weakness of EPN neural network is handling objects with symmetry due to its equivariant design. This weakness has been well explained in the original paper. Due to its own limitation, EPN doesn’t perform well on symmetric objects like bottles compared to KPConv.
>
> #### Q3:  quantitative ablations or examples on limitations
>
> Here we add quantitative ablation showing how well the model will perform after being trained under biased viewpoints distribution. Specifically, we re-generate partial airplane dataset with viwpoints randomly sampled from a 1/4 sphere surface, which indeed hurts the quality of the estimated pose but superisingly not that much. Our method still predicts reasonable poses for those unseen viewpoints and novel instances, as shown below. We do find that the reconstructed shape is not complete for some cases, like the bottom part of the airplane is missing. We will add visualizations showing these limitations into our paper.
>
> <center>Quantitative ablations for model limitation under bias viewpoints distribution</center>
>
> | Dataset           | view points   | Mean R_err | Median R_err | Mean T_err | Median T_err | 5deg. Acc. | 5deg.0.05 Acc. |
> | :---------------- | :------------ | :--------- | :----------- | :--------- | :----------- | :------------ | :---------- |
> | Partial airplane | Unbiased       | **3.47**   | **1.58**     | **0.02**   | **0.02**                                | **0.95**          | **0.88**         |
> | Partial airplane | Biased         | 5.63       | 2.14         | 0.03       | **0.02**                                | 0.91          | 0.83        |
>
> #### Q4: line 117: "only one network" or "the only network"?
>
> EPN is the only equivariant neural network we found that could handle well input point clouds with different sampling patterns, shape variance, and different levels of partialness. While other equivariant networks like SE(3)-transformer also preserves equivariance, it doesn't learn well with variant inputs. We will update the text to make this more precise.

---

### Official Review · Reviewer_ojV5 · 2021-07-16

**Rating:** 6
**Confidence:** 5

**Summary:**

This paper introduces a self-supervised category-level 6D pose estimation to minimize ideally canonical reconstructed point cloud and input point cloud based on rotation-equivariant features. Experiment results can verify the effectiveness on both complete and partial point-based observations.

**Limitations And Societal Impact:**

The following items need to be addressed:
1.  clarify main contribution of the paper;
2. ablation studies on effects of quality of reconstructed shape on pose estimation


**Main Review:**

The paper is difficult to digest due to the writing due to a large number of typos. For example, on lines 65-68, features for 6D pose estimation can be equivariant rather than the problem itself.

Technically, the design of canonical reconstruction and using residual pose predictions for progressive alignment to canonical space is not novel in the field. In the context of category-level 6D pose estimation, the proposed method can be considered to combine two branches of canonical reconstruction and pose estimation in a self-supervised manner. However, due to lack of canonical reference, ideally canonical reconstruction can be affected by intra-class shape variations and inaccurate pose predictions. The authors are suggested to explain its effects on pose estimation and do some ablation studies, which are missing in the paper.



**Time Spent Reviewing:**

3

---

> ### Author Response · Authors · 2021-08-10
> **Response to Reviewer ojV5**
>
> We thank the reviewer for the constructive comments. We will do a careful proofreading and fix all the typos. For example, for lines 65-68, we modify it as ‘the estimated 6D pose should naturally be equivariant with…’;
>
> #### Q1: clarify main contribution of the paper
> * **[Misunderstanding of the contributions]** We are well aware of the reviewer’s concerns on intra-category shape variations and poor canonical shape reconstruction, and that is exactly where our novelty and main contribution come from--by proposing leveraging SE(3) equivariance and designing an effective framework. And our method doesn’t use residual pose predictions for progressive alignment, but could get accurate pose in a single forward pass during the inference stage. We don’t claim novelty over the branch design for canonical reconstruction and pose regression.
> * **[Contribution 1]** We are the first one to explore self-supervised category-level 6D pose estimation from point clouds when there are no CAD models available, allowing intra-category shape variations, and occlusions-introduced partialness, no need of iterative alignment, and handling both symmetric and non-symmetric categories;
> * **[Contribution 2]** We propose the key idea of using networks preserving SE(3) equivariance, which could disentangle pose and shape effectively, and solve this problem elegantly;
> * **[Contribution 3]** Our proposed framework can achieve accurate pose estimation results that are comparable or even beat supervised SOTA methods, for both complete and partial point cloud input in a single forward pass. The secret is that by using rotational-invariant features, our model could learn to predict aligned shape reconstruction in canonical space for different instances through pure end-to-end self-supervised training, and by using rotation-equivariant features on divided SO(3) groups, our pose branch can give accurate SO(3) regression and associated translation;
>
> #### Q2: ablation studies on effects of quality of reconstructed shape on pose estimation
> The reconstructed shape indeed affects pose estimation largely and this is exactly what motivates us to leverage equivariant neural networks, through which we demonstrate for the first time that self-supervised pose estimation methods are able to achieve comparable performance with supervised methods.
>
> * **[Equivariance is the key for high quality aligned reconstruction]** In our major experiments on both complete and partial input, we show that our reconstructed canonical shapes are naturally aligned well against intra-category variations. Also for categories like symmetric bottle, the predicted pose is still quite accurate even the reconstructed shape is not complete(in poor quality), check further visualization in figure 3 and figure 4 in our supp.;
> * **[Ablation studies on reconstruction qualities]** If using general non-equivariant neural networks(like KPConv) for shape prediction, we won’t be able to get category-level aligned canonical shape reconstructions, which would largely affect the accuracy of pose estimation, we list our ablation studies for shape backbone here. Figure 5 in our supplementary also gives a better visualization for drifted canonical shape reconstruction if using KPConv.
> <center>Different Backbones for Canonical Shape Reconstruction</center>
>
> | Dataset           | Shape Backbone | Pose Backbone | Mean R_err | Median R_err | Mean T_err | Median T_err                        | 5deg. Acc. | 5deg.0.05 Acc. |
> | :---------------- | :------------- | :------------ | :--------- | :----------- | :--------- | :---------------------------------- | :------------ | :---------- |
> | Complete airplane | EPN            | EPN           | **23.09**  | **1.66**     | /          | /                                   | **0.87**          | /           |
> | Complete airplane | KPConv         | EPN           | 133.89     | 169.96       | /          | /                                   | 0.00          | /           |
> | Partial airplane  | EPN            | EPN           | **3.47**   | **1.58**     | **0.02**       | **0.02**                                | **0.95**          | **0.88**       |
> | partial airplane  | KPConv         | EPN           | 134.34     | 174.98       | 0.07       | 0.08                                | 0.05          | 0.05        |

---

> > ### Comment · Reviewer_ojV5 · 2021-08-16
> > **Reviewer's response**
> >
> > Thanks for the response. The authors claimed that ``Equivariance is the key for high quality aligned reconstruction'' in the experimental perspective, but it remains unclear that why rotation equivariance can be favorable for canonical shape reconstruction. Intuitively, rotation invariance could be superior to rotation equivariance for shape reconstruction. Moreover, have rotation equivariant feature encoding adopted in 3D object reconstruction to support the authors' claim?

---

> > > ### Author Response · Authors · 2021-08-17
> > > **Clarification**
> > >
> > > Thanks for your follow-up question. We here want to clarify that our paper did exactly what you suggested: we use invariant features for shape reconstruction (see **SE(3)-invariant Canonical Shape Reconstruction Branch**, line 186-217). To be more specific, our proposed method leverages SE(3) invariant features for canonical reconstruction and (typical) SE(3) equivariant feature for pose estimation:
> > > - SE(3)-invariant feature enables disentangled canonical shape reconstruction and encourages category-level consistent alignment;
> > > - SE(3)-(typical)equivariant feature allows highly accurate pose estimation, which also helps the convergence of shape during self-supervised training.
> > >
> > > However, using invariant features doesn't mean that our claim, `equivariance is the key for high quality aligned reconstruction`, is incorrect. The reason is that both "SE(3)-invariance" and "SE(3) equivariance" are equivariance.
> > > This apparent confusion might originate from the definition of equivariance. In our paper (line 56 - 59), we introduced the rigorous definition of equivariance (our definition is the same to that in [Tensor field networks: Rotation- and translation-equivariant neural
> > > networks for 3D point clouds](https://arxiv.org/pdf/1802.08219.pdf)). We can see that invariance is a special case of equivariance for which the equivariant transformation $S_A$ is identical mapping (refer to line 58-59). On the other hand, when $S_A = T_A$ (i.e. the rotation transformation), this type of equivariance is what people may think as the typical rotation equivariance, i.e. the output rotates accordingly when the input rotates. In short, both rotation invariance and (typical) rotation equivariance are equivariance, therefore our claim is at least mathematically correct.
> > >
> > > Furthermore, high quality aligned reconstruction requires pose estimation to be equivariant under our self-supervised setting, i.e. if the pose estimation is not (typically) equivariant with rotation and translation, the canonical reconstruction will also fail. We therefore summarize the request from SE3-invariance in the reconstruction and SE3-(typical)equivariance in the pose estimation as equivariance and say `equivariance is the key for high quality aligned reconstruction`.
> > >
> > > To conclude, the term, `equivariance`, can represent both invariance and (typical) equivariance. That's why we only mention equivariance in the title of our paper and the aforementioned claim. Note that this is a typical choice in many papers, e.g. [SE(3)-Transformers: 3D Roto-Translation Equivariant Attention Networks](https://arxiv.org/abs/2006.10503) can get both SE(3)-invariant and SE(3)-(typical)equivariant features but they only call their network "3D Roto-Translation Equivariant Attention Networks".

---

> > > > ### Author Response · Authors · 2021-08-20
> > > > **Looking Forward to Seeing Your Response**
> > > >
> > > > Dear Reviewer ojV5,
> > > >
> > > > Hopefully our previous clarification has addressed your concern. I believe there is no difference between your suggested reconstruction method and ours. If you think the text of this part is unclear, we will improve the writing by further elaborating the definition of Equivariance in our paper. If you have more questions or concerns, please let us know.
> > > >
> > > > We are looking forward to seeing your further comments or having your supports for accepting our paper.

---

> > > > > ### Comment · Reviewer_ojV5 · 2021-08-22
> > > > > **The authors' reply has addressed all my concern.**
> > > > >
> > > > > The authors' reply has addressed all my concern.

---

### Decision · Program_Chairs · 2021-09-27

**Decision:**

Accept (Poster)

**Comment:**


This paper proposes a self-supervised category-level 6D pose estimation method in which canonical shapes for the input objects are predicted without explicit supervision of orientation. It introduces an  equivariant network architecture and shows its importance for canonical shape estimation.  Reviewers point out missing citations of previous works that consider similar setup (yet different neural architectures), specifically:
[1] Unsupervised Learning of Shape and Pose with Differentiable Point Clouds
[2] Multi-view Consistency as Supervisory Signal for Learning Shape and Pose Prediction
[3] Weakly-supervised 3d shape completion in the wild
and the authors are encouraged to reference these and discuss in detail the difference of their method against the above. Reviewers agree regarding the good empirical performance of the method and the detailed ablations contained in the paper, and the paper is suggested for publication.